# Improving Foundation Model Group Robustness with Auxiliary Sentence Embeddings

**Sisuo Lyu**                                        *sisuolyu@outlook.com*
*The Hong Kong University of Science and Technology (Guangzhou)*

**Hong Liu**[*]                                       *hlynn@xmu.edu.cn*
*Xiamen University*

**Jie Li**                                            *lijie.32@outlook.com*
*Shanghai Artificial Intelligence Laboratory*

**Yan Teng**                                          *tengyan@pjlab.org.cn*
*Shanghai Artificial Intelligence Laboratory*

**Yingchun Wang**                                     *wangyingchun@pjlab.org.cn*
*Shanghai Artificial Intelligence Laboratory*

**Reviewed on OpenReview:** *https://openreview.net/forum?id=5rMtiB96cg*

## Abstract

This paper addresses the critical challenge of mitigating group-based biases in vision-language foundation models, a pressing issue for ensuring trustworthy AI deployment. We introduce DoubleCCA, a novel and computationally efficient framework that systematically enriches textual representations to enhance group robustness. Our key innovation is to leverage an auxiliary large sentence embedding model to capture diverse semantic perspectives, counteracting biased representations induced by limited training data. To this end, we propose a two-stage Canonical Correlation Analysis (DoubleCCA) technique: first, aligning augmented and original embeddings in a shared space; second, reconstructing invariant features to align with visual representations, thus enhancing the model's group robustness. We further propose a simple sentence augmentation approach that aims to improve the robustness of CCA-induced subspaces. Our method is simple to implement and can be easily integrated into existing models, making it a practical solution for improving the robustness of vision-language foundation models to group-based biases. The experiments on a variety of datasets demonstrate that our method outperforms existing methods in terms of both performance and robustness. Our code is available at `https://github.com/sisuolv/doublecca`.

## 1 Introduction

Recently, contrastive language-image pretraining (CLIP) and its variants (Radford et al., 2021; Zhai et al., 2023; Desai et al., 2023) are widely used vision-language models (VLMs). They usually train models on large-scale datasets with a large number of image-text pairs, such as LAION-400M (Schuhmann et al., 2021). Recent works have shown impressive zero-shot generalization on a wide range of tasks, such as medical image classification (Wang et al., 2022), object detection (Ramaswamy et al., 2024), and semantic segmentation (Sun et al., 2024; Li et al., 2024).

However, recent works (Menon & Vondrick, 2022; Roth et al., 2023; An et al., 2024) show that current VLMs lack a systematic investigation of the prompts they use. Therefore, they propose modifying the prompts

---
[*]Corresponding author

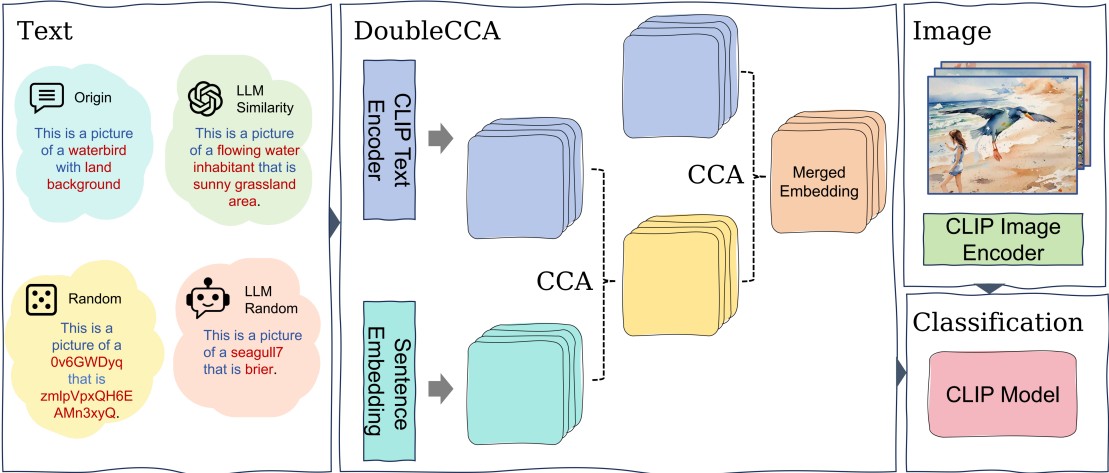

Figure 1: The pipeline of our DoubleCCA. We leverage extra-textual information to augment semantic descriptions and introduce an additional sentence embedding model to complement the semantic limitations of the original VLM text encoder. We use the classical CCA technique twice to merge different semantic information, which helps to improve the group robustness of the CLIP model.

to improve the model's performance, especially its domain generalization ability. Despite their remarkable zero-shot capability, these models are still sensitive to group-based biases, which are attributes correlated with the ground-truth labels but are not directly related to the classification task (Zhang et al., 2024; Dehdashtian et al., 2024a; Zhu & Zhang, 2025).

A robust classifier should be invariant to spurious correlations, *i.e.*, features that are correlated with the ground-truth labels but are irrelevant to the task, such as group attributes. To this end, numerous debiasing methods have been proposed to enhance group robustness (Zhang & Ré, 2022; Kumar et al., 2022; Kirichenko et al., 2023; Chuang et al., 2023; Dehdashtian et al., 2024c; You et al., 2024; Gao et al., 2024; Phan et al., 2024; Yang et al., 2024). Many of these approaches involve training a lightweight adapter on top of a frozen CLIP model, using data annotated with both target and group labels. However, despite their success, these methods often suffer from critical limitations that we aim to address.

First, the performance of the model is highly dependent on the dataset used for training the newly added adapter architecture, which hinders the model's ability to generalize efficiently to other datasets. Second, other works (Chuang et al., 2023; Yang et al., 2024) employ prompt tuning, which often relies on externally constructed knowledge, frequently generated using LLMs. For instance, Yang et al. (2024) uses an LLM to synthesize a balanced textual dataset and then optimizes prompts via fine-tuning to improve robustness. This reliance on LLMs to construct complex training sets makes it difficult for models to generalize quickly to other datasets. Moreover, some prompt tuning methods may incur additional API costs, which are not efficient. Therefore, current debiasing methods exhibit limitations in generalization and efficiency.

To address these challenges, we ask the following question: *How can we improve the group robustness of foundation models without relying on prior knowledge of the dataset?* To answer this question, we introduce DoubleCCA, a novel framework for enhancing the group robustness of vision-language foundation models (*e.g.*, CLIP model) against group-based biases. Our approach is motivated by the observation that CLIP's text encoder has a limited capacity to capture rich semantic information, which can lead to biased representations. Thus, our key idea is to leverage an auxiliary sentence embedding model to generate semantically richer text embeddings, thereby complementing the limitations of the original CLIP text encoder.

Specifically, for a given set of class descriptions, we generate two distinct sets of text embeddings: one from CLIP's text encoder and another from an auxiliary sentence embedding model. We then introduce a two-stage Canonical Correlation Analysis (CCA) framework. The first stage aligns these two embedding sets into a shared, semantically correlated space. The second stage merges these aligned representations and projects the result back into CLIP's original embedding space to ensure compatibility with the visual features. However, a critical challenge arises when the number of classes is small, as is common in many datasets. This

provides insufficient data for CCA to learn stable transformation matrices. To address this, we propose a data augmentation scheme to generate a set of diverse sentence embeddings, thereby enabling a more robust estimation of the transformations. Note that the proposed data augmentation is not intended to completely avoid using LLMs, but rather to minimize heavy reliance on interaction with LLMs, such as LLM-based synthesized datasets or dataset-specific prompt optimization.

The pipeline of the entire framework is shown in Figure 1, and our contributions are summarized as follows:

- We propose a novel method, called DoubleCCA, to improve the group robustness of foundation models against group-based biases.
- We introduce an additional sentence embedding model to complement the semantic limitations of the original CLIP text encoder through the CCA technique.
- We demonstrate the effectiveness of our method on a variety of datasets, where it outperforms existing methods in terms of both group robustness and domain generalization.

## 2   Preliminaries

This section will introduce the necessary background knowledge for our method, including the CLIP foundation model and Canonical Correlation Analysis (CCA).

**CLIP model.** The CLIP model (Radford et al., 2021) is a vision-language foundation model that consists of two parts: a vision encoder and a text encoder. The vision encoder $\Phi_v : \mathbb{R}^{d_v} \to \mathbb{R}^d$ and the text encoder $\Phi_t : \mathbb{R}^{d_t} \to \mathbb{R}^d$ are deep models that map the input image and text to a $d$-dimensional embedding space, respectively. Given a batch of image-text pairs $(I, T)$, the model is trained to minimize the symmetric contrastive loss Radford et al. (2021), which aligns the image-text embedding pairs in the representation space $\mathbb{R}^d$.

Once the model is trained, we can directly use the image and text encoders to align images with text descriptions. Thus, a zero-shot image classifier can be built by comparing the similarity between the image embedding $\Phi_v(I)$ and the text embedding $\Phi_t(T)$. The typical method is to combine the name of the class $k$ with the predefined template to obtain the text description $t_k$. For example, the class of zebra can be integrated into the prompt template "a photo of a ⟨class name⟩" to yield the description "a photo of a zebra". Thus, we can compute the logits for each class by using the cosine similarity between the image embedding and the text embedding, and the class with the highest score is the predicted class.

**Group Robustness Metrics.** For a model $f : \mathcal{X} \to \mathcal{Y}$, we define the group-specific accuracy as:

$$\text{Acc}_{y,g}(f) = \mathbb{E}_{x \sim \mathcal{D}_{y,g}}[\mathbf{1}(f(x) = y)], \tag{1}$$

where $Acc_{y,g}$ is the accuracy of class $y$ that belongs to group $g$. The worst group accuracy, which measures the model's robustness to spurious correlations, is defined as:

$$\text{Acc}_{\text{worst}}(f) = \min_{(y,g) \in \mathcal{Y} \times \mathcal{G}} \text{Acc}_{y,g}(f). \tag{2}$$

A model with high average accuracy but low worst-group accuracy indicates susceptibility to spurious correlations. The robustness gap is defined as:

$$\text{Gap}(f) = \mathbb{E}_{(x,y) \sim \mathcal{D}}[\mathbf{1}(f(x) = y)] - \text{Acc}_{\text{worst}}(f). \tag{3}$$

**Canonical Correlation Analysis (CCA).** Canonical Correlation Analysis (CCA) is a statistical method that finds the transformation that maximizes the correlation between two feature sets from different models. Let $X_A \in \mathbb{R}^{n \times d_A}$ and $X_B \in \mathbb{R}^{n \times d_B}$ be the data matrices, where $n$ is the number of samples, and $d_A$ and $d_B$ are the dimensions of the feature vectors. CCA finds the transformation matrices $W_A$ and $W_B$ that maximize the correlation between the transformed features $Z_A = X_A W_A$ and $Z_B = X_B W_B$ in a common feature space.

Figure 2: We compare the performance of different prompts with different backbone models on the Waterbirds dataset. "Ori" denotes the original prompt of CLIP, *i.e.*, "a photo of a ⟨class name⟩". "Waffle-1" denotes the combination of the original prompt and the random words, *i.e.*, "a photo of a ⟨class name⟩, which has ⟨random word⟩". "Waffle-2" also denotes the combination of the original prompt and the random words, but with a different template, *i.e.*, "a photo of a ⟨class name⟩, ⟨random characters⟩".

We further define $S^{XX} = X_A^T X_A$ and $S^{YY} = X_B^T X_B$ as the covariance matrices of $X_A$ and $X_B$, and $S^{XY} = X_A^T X_B$ as the cross-covariance matrix. Therefore, the formulation of CCA can be written as follows:

$$\max_{W_A, W_B} \quad \text{corr}(Z_A, Z_B) = W_A^T S^{XY} W_B$$
$$\text{s.t.} \quad W_A^T S^{XX} W_A = I, \quad W_B^T S^{YY} W_B = I, \tag{4}$$

where $\text{corr}(Z_A, Z_B)$ is the correlation between $Z_A$ and $Z_B$, and $I$ is the identity matrix.

This formulation can be solved by the eigenvalue decomposition of the generalized eigenvalue problem:

$$U, S, V^T = SVD\big((S^{XX})^{-1/2} \cdot S^{XY} \cdot (S^{YY})^{-1/2}\big),$$
$$W_A = (S^{XX})^{-1/2}U, \quad W_B = (S^{YY})^{-1/2}V.$$

In practice, we center the data before applying CCA to ensure that the data has a mean of zero. And we use regularized CCA (Corrochano et al., 2005; Horoi et al., 2024) to make the computation of $W_A$ and $W_B$ more stable.

**Sentence Embedding Models.** Sentence embedding models map variable-length sentences to fixed-dimensional dense vectors that capture semantic meaning. Unlike CLIP's text encoder, which is optimized for vision-language alignment, dedicated sentence embedding models like Sentence-BERT (Reimers & Gurevych, 2019), HiT (He et al., 2024), BART (Lewis et al., 2020), L12-V2(Reimers & Gurevych, 2020a), and GTE (Li et al., 2023) are trained specifically for semantic similarity tasks. These models learn embedding features where semantically similar sentences are mapped to nearby points in the embedding space, providing complementary information to CLIP's vision-oriented text representations.

## 3 Method

### 3.1 Problem Analysis

One interesting approach to improving CLIP's zero-shot classification is to augment the prompts with additional visual concepts from external knowledge sources. Menon & Vondrick (2022) utilizes large language models (LLMs) like GPT-3 to generate class-specific descriptions for each class and incorporate them into prompts, resulting in prompts like "a photo of a hen, which has two legs." However, this method is limited by prior knowledge of the class name, and the GPT-3-generated descriptions exhibit a high degree of ambiguity and limited visual relevance.

Roth et al. (2023) propose a method called WaffleCLIP, which substitutes GPT-3 generated descriptors with random word or character sequences, resulting in prompts such as "a photo of a hen, which has jmhj, !J#m." Where "jmhj, !J#m" is the random character sequences. Based on WaffleCLIP, we simply study the effect of this method on the group robustness of the CLIP model. We conduct four toy experiments on the Waterbirds dataset (Sagawa et al., 2020) with four different backbone models, *i.e.*, ResNet-50, ViT-B/32, ViT-B/16, and

ViT-L/14. We compare the results of vanilla CLIP with the original prompt, WaffleCLIP with random words (shortened as Waffle-1), and WaffleCLIP with random characters (shortened as Waffle-2). See Figure 2.

We observe that WaffleCLIP methods achieve better results in terms of average accuracy and worst-group robustness only when using the ViT-L/14 backbone. However, for the other three backbone models, their performance is worse than that of the vanilla CLIP. Moreover, when using ViT-B/16 or ResNet-50 as the backbone, WaffleCLIP's worst-group robustness drops to near zero, which is substantially lower than that of the original prompt. In other words, WaffleCLIP exhibits inconsistent performance on debiasing tasks, which stands in notable contrast to the empirical findings documented in its original publication.

Although WaffleCLIP enhances semantic representations by incorporating stochastic words, it exhibits persistent deficiencies in semantic enhancement capacity – particularly evident in debiasing tasks. On the other hand, PerceptionCLIP's two-stage paradigm first predicts attribute-specific weighting priors before final category determination. While its empirical results reported in (An et al., 2024) show non-trivial debiasing capabilities, substantial performance gaps persist when benchmarked against state-of-the-art alternatives.

Since WaffleCLIP and PerceptionCLIP do not modify the CLIP text encoder, we believe their suboptimal performance is due to the text encoder's failure to produce more semantically meaningful text embeddings. Consequently, we investigate the use of auxiliary text embedding models from natural language processing, which are designed to generate more informative text representations. We further perform an empirical analysis of CLIP's text representations to support this claim.

Figure 4 illustrates the feature distributions of various sentence embedding models (including the CLIP text encoder, Sentence-BERT, BART, and L12-V2) visualized using t-SNE (van der Maaten & Hinton, 2008) on the Waterbirds dataset, which is widely used for evaluating group robustness. The CLIP text encoder exhibits long arrows between centroids, indicating significant within-class shifts due to spurious background correlations. Conversely, models like BERT and L12-V2 display markedly shorter arrows, demonstrating more invariant representations.

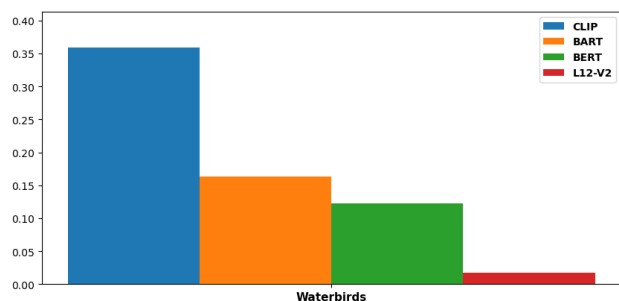

Moreover, we show the quantitative results on Waterbirds in Figure 3. The CLIP text encoder exhibits a noticeable attribute bias of 0.36, while BERT significantly reduces this bias to 0.12. A similar trend is observed for the other sentence embedding models tested, and preliminary results suggest that this trend also holds for other benchmarks, such as CelebA (see Appendix for details). These results

Figure 3: Quantitative comparison of attribute bias across text encoders on Waterbirds datasets. Attribute bias measures the average L2 distance between class-conditional attribute centroids in the embedding space (see Appendix B for details). Lower values indicate more invariant representations.

support our claim that auxiliary sentence embedding models are beneficial for extracting more invariant representations. A detailed theoretical motivation for this claim is provided in the Appendix. Therefore, we introduce DoubleCCA, a method that leverages an auxiliary model to effectively enhance the foundation model's performance and group robustness. We will detail our method in the following section.

## 3.2 DoubleCCA

Based on the previous analysis, our main idea is to utilize these sentence embedding models to enrich the text embeddings of the CLIP model. However, there are two major challenges in this process. First, the dimensionality of the text embeddings generated by the sentence embedding model may not be the same as that generated by the CLIP text encoder. Second, it is difficult to merge these newly generated sentence embeddings into the CLIP model. To address these challenges, we propose a novel method called DoubleCCA, which utilizes the canonical correlation analysis (CCA) technique twice. The first CCA is used to align the representations of different embeddings into a common space. The second CCA is used to merge the aligned representations and then recover to the original embedding space.

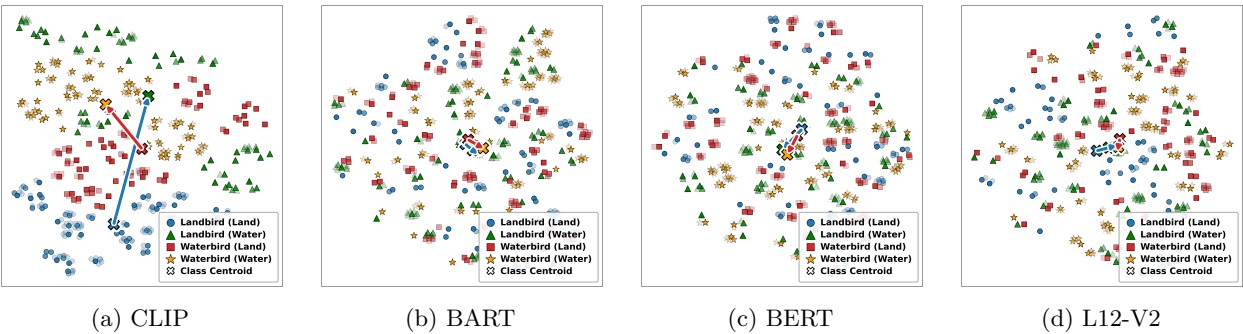

|            |            |            |            |
| :--------: | :--------: | :--------: | :--------: |
| (a) CLIP   | (b) BART   | (c) BERT   | (d) L12-V2 |

Figure 4: Group-bias visualization of text embeddings across text encoders on the Waterbirds dataset. Each panel shows 2D t-SNE projections of text embeddings from different encoders: (a) CLIP, (b) BART, (c) BERT, (d) L12-V2. Points represent individual text embeddings, with colors indicating class-background combinations. Arrows connect centroids of the same class across different backgrounds (e.g., Landbird on land $\rightarrow$ Landbird on water). Shorter arrows indicate smaller spurious attribute shifts, demonstrating that auxiliary text encoders (BART, BERT, L12-V2) produce more invariant representations compared to CLIP.

### 3.2.1 Step 1: The First CCA

We first generate sentence embeddings using the sentence embedding model $\Phi_{se}$ and the CLIP text encoder $\Phi_t$. Let $X \in \mathbb{R}^{n \times d}$ and $X_{se} \in \mathbb{R}^{n \times d_{se}}$ be the data matrices, where $n$ is the number of classes in the dataset, $d$ and $d_{se}$ are the dimensions of the text embeddings generated by the CLIP text encoder and the sentence embedding model, respectively. We then apply CCA (w.r.t. Eq.4) to learn the transformation matrices $W_x$ and $W_{se}$ that embed two features into a common space:

$$Z_x = XW_x, \quad Z_{se} = X_{se}W_{se}, \tag{5}$$

where $Z \in \mathbb{R}^{n \times d_{cca}}$ and $Z_{se} \in \mathbb{R}^{n \times d_{cca}}$ are the aligned representations of the sentence embeddings and the CLIP text embeddings, respectively.

### 3.2.2 Step 2: The Second CCA.

In zero-shot classification, CLIP computes the similarity between image and text embeddings: $S(I, y) = f_t^T f_v$, where $f_v = \Phi_v(I)$ and $f_t = \Phi_t(T_y)$ for class $y$. After the First CCA, we achieved two different scores:

$$S_x(I, y) = x^{(y)^T} W_x W_x^T f_v, \ S_{se}(I, y) = x_{se}^{(y)^T} W_{se} W_x^T f_v, \tag{6}$$

where $x^{(y)}$ and $x_{se}^{(y)}$ are the text embeddings of the class $y$ w.r.t. the original prompts.

**Optimal Merging Strategy.** To combine these complementary scores effectively, we formulate the merging as an optimization problem. Let $\hat{W}_x = XW_xW_x^T$ and $\hat{W}_{se} = X_{se}W_{se}W_x^T$ represent the projected text embeddings.

*Intuition:* The optimal linear combination that maximizes robustness to group shifts while preserving classification accuracy can be achieved through a second CCA that aligns the two predictor spaces.

Following (Horoi et al., 2024), we apply CCA to merge these predictors. First, we construct proxy features:

$$X_A = \hat{W}_x X, \quad X_B = \hat{W}_{se} X, \tag{7}$$

where we use the original text embeddings $X$ as a proxy for image features. We then solve:

$$\max_{P_A, P_B} \quad \mathrm{tr}(P_A^T S_{AB} P_B) \quad \text{s.t. } P_A^T S_{AA} P_A = I, \quad P_B^T S_{BB} P_B = I, \tag{8}$$

where $S_{AA} = X_A^T X_A$, $S_{BB} = X_B^T X_B$, and $S_{AB} = X_A^T X_B$.

---

**Algorithm 1** DoubleCCA

---

**Require:** Sentence embedding model $f_{se}$, CLIP model $(f_v, f_t)$, number of augmented sentences $K$
**Ensure:** Merged text embeddings $W$
 1: Generate $K$ augmented sentences for each class
 2: Extract sentence embeddings $F_{rse}$ using $\Phi_{se}$
 3: Extract CLIP text embeddings $F_r$ using $\Phi_t$
 4: Apply CCA to $X$ and $Y$ to obtain $W_x$ and $W_{se}$
 5: Compute $\hat{W}_x = XW_xW_x^T$, $\hat{W}_{se} = X_{se}W_{se}W_x^T$
 6: Generate augmented sentence embedding features $F_r$
 7: Compute $X_A = \hat{W}_xF_r$, $X_B = \hat{W}_{se}F_r$
 8: Apply CCA to $X_A$ and $X_B$ to obtain $P_A$ and $P_B$
 9: Compute $M = (P_B \cdot P_A^{-1})^T$
10: Merge text embeddings: $W = \frac{1}{2}(\hat{W}_x + M \cdot \hat{W}_{se})$
11: **return** $W$

---

The merged text embeddings are:

$$W = \frac{1}{2}(\hat{W}_x + M \cdot \hat{W}_{se}), \quad M = (P_B \cdot P_A^{-1})^T. \tag{9}$$

This merging preserves semantic coherence from both sources while reducing sensitivity to spurious correlations.

Then, we can apply CCA to learn the transformation matrices $P_A$ and $P_B$ by maximizing the correlation between $X_A$ and $X_B$ as follows:

$$\max_{P_A, P_B} \quad \text{corr}(X_A, X_B) = P_A^T S^{AB} P_B \quad \text{s.t.} \quad P_A^T S^{AA} P_A = I, \quad P_B^T S^{BB} P_B = I,$$
$$S^{AA} = X_A^T X_A, \quad S^{BB} = X_B^T X_B, \quad S^{AB} = X_A^T X_B. \tag{10}$$

### 3.2.3 Data Augmentation for Stable CCA

We note that the number of class labels is usually much smaller. For example, there are only two classes in the Waterbirds dataset. This means that only two sentences are used for the CCA to learn the transformation matrices $W_x$ and $W_{se}$. We think this is not enough to learn stable transformation matrices. (The next section will show the experimental verifications.) To address this issue, we propose using data augmentation to generate more sentence embeddings. First, we combine the original prompt and the random character sequences, *i.e.*, "a photo of a ⟨class name⟩, ⟨random sequences⟩". We call this *random sentence*. Then, we use a large language model (like Qwen) to infer more complementary information that is similar to the original prompt. Third, we concatenate two types of sentences with the original sentences to form a new sentence set, which has a size of K. Finally, we use the sentence embedding model and the CLIP text encoder to extract the corresponding sentence embedding features, *i.e.*, $F_{rse}$, and $F_r$ respectively. We replace $X_{se}$ with $F_{rse}$ and $X$ with $F_r$ to apply CCA to learn the transformation matrices $W_x$ and $W_{se}$.

### 3.2.4 Inference

After DoubleCCA, we can achieve the merged text embedding matrix $W \in \mathbb{R}^{n \times d}$. We can directly use these merged text embeddings to predict the class label of the input image, which can be formulated as follows:

$$\hat{y} = \arg\max_{y \in \mathcal{Y}} S(I, y), \text{ where } S(I, y) = W_y \Phi_v(I), \tag{11}$$

where $W_y \in \mathbb{R}^{1 \times d}$ is the $y$-th row of the merged embedding matrix $W$, which is the embedding feature of $y$.

The overall process of DoubleCCA is summarized in Algorithm 1. DoubleCCA is designed as a plug-and-play module, allowing it to be combined with various existing methods such as PerceptionCLIP (PCLIP) (An et al., 2024) and Oth-Cal (Chuang et al., 2023). The specific combination schemes are detailed in the Appendix.

The overall time complexity of our DoubleCCA is: $\mathcal{O}(K(d^2 + d_{se}^2) + d^3)$, where the time complexity of the First CCA is $\mathcal{O}(Kd^2 + Kd_{se}^2 + \min(d, d_{se})^3)$ and the time complexity of the Second CCA is $\mathcal{O}(Kd^2 + d^3)$. In a typical experimental setting ($e.g. K \approx 500$, $d = 512$ for ViT-B), DoubleCCA introduces negligible overhead to standard CLIP inference. Since the optimization is convex, it can be solved with eigenvalue decomposition to get a closed-form solution; it's guaranteed to find the best solution and converge reliably.

Table 1: Average accuracy and worst-group robustness on the Waterbirds and CelebA datasets. We compare our method with original CLIP and recent PerceptionCLIP, and we select four backbones: ResNet-50, ViT-B/32, ViT-B/16, and ViT-L/14.

| | | RN50 | | | ViT-B/32 | | | ViT-B/16 | | | ViT-L/14 | | |
| | | avg.↑ | worst↑ | gap↓ | avg.↑ | worst↑ | gap↓ | avg.↑ | worst↑ | gap↓ | avg.↑ | worst↑ | gap↓ |
|---|---|---|---|---|---|---|---|---|---|---|---|---|---|
| Waterbirds | CLIP | 90.47 | 16.07 | 74.40 | 87.34 | 47.28 | 40.06 | **87.34** | 26.79 | 60.55 | 90.55 | 44.64 | 45.91 |
| | +background | 90.62 | 39.29 | 51.33 | 78.58 | 61.96 | **16.62** | 86.01 | 44.34 | 44.73 | 87.72 | 59.98 | 27.74 |
| | Ours | **91.76** | 44.64 | 47.30 | **89.34** | 57.60 | 31.74 | 86.53 | 28.58 | 57.95 | **92.14** | 51.78 | 40.36 |
| | + background | 91.03 | **48.21** | **42.82** | 85.44 | **62.50** | 22.94 | 86.43 | **46.43** | **40.00** | 89.55 | **62.50** | **27.05** |
| CelebA | CLIP | 81.05 | 73.87 | 7.18 | 80.73 | 75.82 | 4.91 | 75.16 | 62.01 | 13.15 | **86.98** | 77.36 | 9.62 |
| | +gender | 85.97 | 81.58 | 4.39 | 80.18 | 76.18 | **4.00** | 75.92 | 66.71 | 7.99 | 80.30 | 74.31 | 5.99 |
| | +gender,age | 87.74 | 84.94 | 2.80 | 82.34 | 77.21 | 5.13 | 75.22 | 64.61 | 10.61 | 82.26 | 79.06 | 3.21 |
| | +gender,age,race | 85.91 | 82.57 | 3.34 | 81.99 | 75.67 | 6.32 | 76.37 | 67.93 | 8.44 | 82.77 | 80.00 | 2.77 |
| | Ours | 85.35 | 83.05 | 2.30 | **84.19** | **78.75** | 5.44 | 79.21 | 68.54 | 10.67 | 85.79 | 81.18 | 4.61 |
| | +gender | 87.53 | 85.56 | 1.97 | 82.67 | 76.87 | 5.80 | **78.55** | **73.84** | **4.71** | 81.44 | 76.14 | 5.30 |
| | +gender,age | **88.70** | **86.35** | 2.35 | 82.16 | 76.90 | 5.44 | 78.09 | 70.54 | 7.55 | 83.78 | 80.87 | 2.91 |
| | +gender,age,race | 85.93 | 84.18 | **1.75** | 82.63 | 75.92 | 6.71 | 77.17 | 69.18 | 7.99 | 85.35 | **83.00** | **2.35** |

# 4 Experiments

## 4.1 Experimental Setup

**Datasets.** We evaluate the group robustness of our method. We conduct experiments on two widely used datasets: Waterbirds (Sagawa et al., 2020) and CelebA (Liu et al., 2015). For these two datasets, each image has an associated group attribute, such as the background of the image in the Waterbirds dataset and the gender/age of the person in the CelebA dataset. All these attributes are correlated with the ground truth labels, but they are not directly related to the classification task. Following previous work (Zhang & Ré, 2022), we consider these attributes as group attributes and report the average accuracy and the worst-group robustness on these datasets. In Appendix G, we also compare our method with the original CLIP on the other five benchmark datasets, including CMNIST (LeCun et al., 2002), FairFace (Karkkainen & Joo, 2021), CounterAnimal (Wang et al., 2024), ImageNet-A (Hendrycks et al., 2021), and COCO-FP (Liu et al., 2024).

**Implementation Details.** We utilize CLIP (Radford et al., 2021) as the foundation model and evaluate the performance of our method on various tasks and datasets. All experiments use PyTorch (Paszke et al., 2019) and are performed on a single NVIDIA A100 GPU. We follow the same experimental settings as previous work (An et al., 2024). We use Resnet-50 (He et al., 2016), ViT-B/32, ViT-B/16, and ViT-L/14 (Dosovitskiy et al., 2021) as the backbone models for the evaluation of group robustness. For the evaluation of domain generalization, we use ViT-B/16 as the backbone model.

For the sentence embedding model, we use the Hierarchy Transformer encoder (HiT) (He et al., 2024) as the default sentence embedding model.[1] Since the output of the HiT lies in the hyperbolic space, we use the logarithmic map function to transform the output to the Euclidean space (Yang et al., 2023). We set the dimension of the common space in the first CCA to 64, and the dimension of the second CCA is set to the dimension of the original image embeddings. Moreover, we set the number of augmented sentences $K$ to 500, which can achieve good results empirically.

---

[1]In our experiments, we use "HiT-MiniLM-L12-WordNetNoun" released on HuggingFace as the sentence embedding model.

Table 2: Comparison with various state-of-the-art methods on the Waterbirds and CelebA datasets. The backbone model is ViT-L/14.

| Method | Waterbird | | | CelebA | | |
|---|---|---|---|---|---|---|
| | avg. | worst | gap | avg. | worst | gap |
| CLIP | 90.55 | 44.64 | 45.64 | **86.98** | 77.36 | 9.62 |
| WaffleCLIP | 91.57 | 57.14 | 34.43 | 84.50 | 79.35 | 5.15 |
| Oth-Cal | 84.71 | 67.13 | 17.58 | 86.19 | 76.11 | 10.08 |
| FairerCLIP | 88.87 | 77.57 | 11.30 | 82.57 | 78.49 | 4.11 |
| PCLIP | 87.72 | 59.98 | 27.74 | 82.77 | 80.00 | 2.77 |
| Ours | **92.14** | 51.78 | 40.36 | 85.79 | 81.18 | 4.61 |
| Ours+PCLIP | 89.55 | 62.50 | 27.05 | 85.35 | 83.00 | 2.35 |
| Ours+Oth-cal | 84.04 | **80.53** | **3.51** | 85.90 | **84.51** | **1.39** |

## 4.2 Results on Group Robustness

We first evaluate the group robustness of our method on the Waterbirds and CelebA datasets. The results are reported in Table 1.[2] We mainly evaluate four different backbone models (RN50, ViT-B/32, ViT-B/16, and ViT-L/14). The results are compared between the baseline CLIP model and our proposed method.

First, we show the results when the text prompts only describe the class and ignore the contextual attributes. First, we observe that our method achieves better average accuracy and worst-group robustness than the baseline CLIP model on both datasets. Although the average accuracy of our method is slightly lower than that of the baseline CLIP model, when the backbone is ViT-B/16 on Waterbirds and the backbone is ViT-L/14 on CelebA, the worst-group robustness is significantly improved. We think this is a trade-off between average accuracy and worst-group robustness, which has also been observed in recent work (Dehdashtian et al., 2024b). For example, when the backbone is ViT-L/14 on CelebA, the worst-group robustness of our method is 81.18%, which is higher than that of the baseline CLIP model (77.36%). However, the average accuracy has a slight decrease (from 86.98% to 85.79%) compared to the baseline CLIP model.

Following PerceptionCLIP (An et al., 2024), we include contextual attributes such as conditional information, background information in the Waterbirds dataset, and gender information (*i.e.*, female and male) in the CelebA dataset. Here, we only substitute the original prompt embedding with the merged text embeddings $W$ in the CLIP model and then use the same inference process as in (An et al., 2024).

We report the results on Waterbirds by considering the background as the contextual attributes, such as in a forest, in the sky, on a street, on grass, on a tree, with flowers, on a beach, with humans, on a branch, etc. First, the same phenomena are observed in our reproduced results, where group robustness can be improved by incorporating these attributes, which also help reduce the accuracy gap and achieve a fairer zero-shot classifier. Second, we observe that by using our method, worst-group robustness can be further improved, and the accuracy gap can be further reduced in most cases. More interestingly, considering the background information, the worst-group robustness shows consistent improvement across different backbone models, while the average accuracy has experienced a slight decrease. In this case, a trade-off between average accuracy and worst-group robustness is also observed. However, we believe this will be beneficial in achieving a fairer zero-shot classifier.

Then, we also report the results on CelebA by considering contextual attributes such as gender (female and male), age (young and old), race (white skin, dark skin, Asian, and others), etc. We observe that our method can achieve an overall better average accuracy and worst-group robustness than the baseline CLIP model on the CelebA dataset. For instance, when the backbone is ViT-B/16, the accuracy gap of our method is 4.71%, which is lower than that of the baseline CLIP model (7.99%), considering the contextual attribute of gender. Furthermore, compared to the results shown in FairerCLIP, our method achieves better results when the backbone is ResNet-50, where the best worst-group robustness of our method is 86.35%, which is higher than that of FairerCLIP (81.50%). For the backbone of ViT-L/14, our method also achieves a competitive result

---

[2]Note that in this table, "+background" on the Waterbirds and "+gender, +gender, age, +gender, age, race" on CelebA refer to the recent PerceptionCLIP method. Our method can be integrated with the PerceptionCLIP method, where we simply replace the original text embeddings with the merged text embeddings, as shown in Eq. 11.

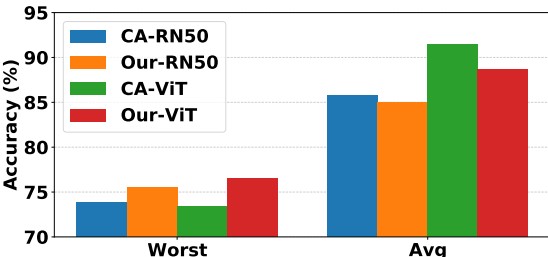

Figure 5: Combination of Contrastive Adapter (CA) and DoubleCCA. We report the average accuracy and worst-group robustness on the Waterbirds dataset. The backbone model is ViT-L/14 and ResNet-50.

compared to FairerCLIP, where the best worst-group robustness of our method is 83.00%, which is slightly lower than that of FairerCLIP (85.20%). It is worth noting that FairerCLIP utilizes the target label and attributes to learn a kernel map function in a supervised way, which is more complex than our method.

Third, we compare our method with various state-of-the-art methods, including WaffleCLIP (Roth et al., 2023), PerceptionCLIP (PCLIP) (An et al., 2024), FairerCLIP (An et al., 2024), and Oth-Cal (Chuang et al., 2023). Moreover, since our method can be easily integrated into existing models, we also combine our method with PCLIP and Oth-Cal to further improve the group robustness of the CLIP model. The results are shown in Table 2. We observe that our method can achieve the overall best results on both datasets. Note that although our method does not achieve the best average accuracy on the CelebA dataset, it and its combination variants achieve higher worst-group robustness. On the other hand, the results on these two benchmarks show that our method indeed helps reduce the gap between the average accuracy and the worst-group robustness. This means that using auxiliary information can help achieve a good trade-off (Dehdashtian et al., 2024b).

Finally, we also combine our method with the contrastive adapter (CA) (Zhang & Ré, 2022) to further improve the group robustness of the CLIP model. In detail, we first use DoubleCCA to generate the merged text embeddings, and then substitute the original text embeddings with the merged text embeddings in the CLIP model. Finally, we use the CA algorithm to learn the adapter. The results are shown in Figure 5. We observe that using the merged text embeddings helps improve the worst-group accuracy, but the average accuracy has decreased slightly. Thus, the trade-off between the average accuracy and the worst-group robustness is also observed.

Overall, the results show that DoubleCCA effectively enhances the group robustness of foundation models, providing better performance and fairness across different datasets and backbone models. In different scenarios, trade-off phenomena are also observed, which is consistent with previous work (Dehdashtian et al., 2024b).

## 4.3 Effect of Sentence Embeddings

Since DoubleCCA leverages auxiliary sentence embeddings, we conduct an ablation study to analyze the effect of sentence embeddings on the group robustness of the CLIP model.

In previous experiments, we used the HiT model (He et al., 2024) to generate sentence embeddings. To further study the effect of sentence embeddings, we replace the HiT model with other sentence embedding models. To ensure a comprehensive comparison, we select popular models from the HuggingFace Hub[3] as alternatives to the default HiT model, such as the classical Sentence-BERT model (BERT) (Reimers & Gurevych, 2020b), gte-base-en-v1.5 model (GTE) (Li et al., 2023), and bart-base model (BART) (Lewis et al., 2020). We directly use the pre-trained models released by the HuggingFace Hub to generate sentence embeddings for the Waterbirds dataset. The results are shown in Figure 6.

Compared with the original CLIP model, we observe that different sentence embedding methods in DoubleCCA either improve the model's performance or maintain it at a comparable level. Notably, HiT demonstrates the

---

[3]https://huggingface.co/

most significant performance improvements. Both Sentence-BERT and gte-base-en-v1.5 also have a positive impact on the model's performance.

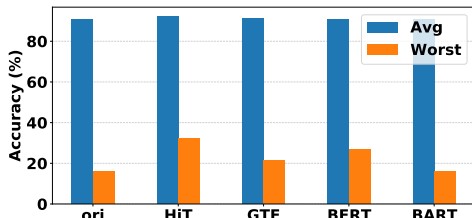

Figure 6: The impact of different sentence embedding models. We conduct experiments on the Waterbirds and report the average accuracy (Avg) and worst-group robustness (WG).

First, HiT is a state-of-the-art sentence embedding model that aims to learn the hierarchical semantic structure in language models. HiT is trained on WordNet, which can provide unseen subsumptions and hypernyms for the words in the sentence. Second, Sentence-BERT and gte-base models are also popular sentence embedding models that have been verified to be effective in unsupervised text retrieval tasks. However, BART shows little improvement in model performance. We believe this is because BART targets dialog understanding, question answering, and summarization tasks, which may face the same problems as mentioned before, where it will introduce semantic ambiguity to text embeddings (Menon & Vondrick, 2022).

Overall, the results demonstrate that the choice of the sentence embedding model can significantly affect the performance of the foundation model. We recommend using HiT as the default sentence embedding model in DoubleCCA, as it achieves the best performance in our experiments. Moreover, it is more interesting to explore the effect of different sentence embedding models on the group robustness of the foundation model, which is left for future work.

### 4.4 Ablation Study

#### 4.4.1 Effect of the Hyperparameters.

**Number of Sentences.** We conduct an ablation study to analyze the effect of the number of augmented sentences on group robustness. We employ the backbone model for ResNet-50 and fix the dimension of the CCA at 64. Then, we vary the number of sentences from 1 to 2000. The results are shown in Figure 7 (a). The results indicate that varying the number of sentences has minimal impact on the average accuracy but demonstrates a substantial influence on worst-group robustness. When the number of sentences is less than 500, the worst-group robustness exhibits high variability. In particular, when the number of sentences drops to 100, the performance deteriorates below that of the original CLIP model. We attribute this instability to the inherent randomness of the part of random sentences. However, as the number of sentences increases, the model's performance gradually stabilizes, suggesting that sufficient sentences enable the model to capture meaningful pragmatic information.

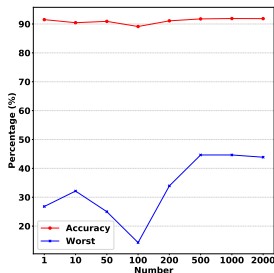
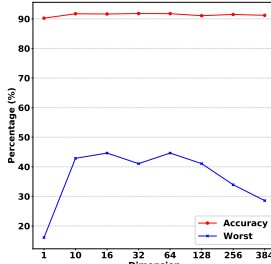
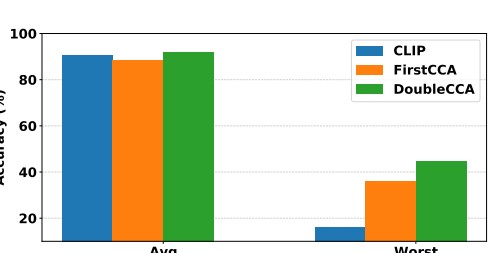

(a) # of augmented sentences     (b) Dimension of CCA     (c) Effect of the first CCA

Figure 7: Ablation study results on the Waterbirds dataset.

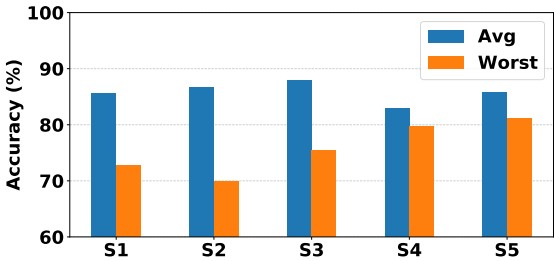

Figure 8: Accuracy of ViT-L/14 on CelebA with five different data augmentation methods (more details can be seen in Sec. 4.4.3). We report the average accuracy (Avg) and worst-group robustness (WG).

**Dimension of CCA.** We further study the effect of the dimension of the CCA on the group robustness of the CLIP model. We employ the ResNet-50 backbone model and fix the number of sentences to 500. Then, we vary the dimension of the CCA from 1 to 384[4]. The results are shown in Figure 7 (b). The results indicate that the dimension of the common space significantly impacts performance. Both low and high dimensions adversely affect the results; low dimensions lead to insufficient feature representation, while high dimensions introduce feature vectors corresponding to small singular values. We recommend setting the dimension of the CCA to 64, as it achieves the best performance in our experiments. Moreover, as discussed in (Vidal et al., 2016), the dimension of this subspace is a natural measure of the model complexity; thus, some automatic dimension selection methods can be used to determine the optimal dimension of the CCA. We leave this for future work.

### 4.4.2 Effect of the First CCA

Finally, we analyze the effect of the first CCA on the group robustness of the CLIP model. We employ the backbone model of ResNet-50 and fix the number of sentences to 500. Then, we remove the second CCA from the DoubleCCA method and directly use Eq.6 as the score function for the zero-shot classification. The results are shown in Figure 7 (c). The results indicate that only the first CCA has a positive impact on the group robustness of the CLIP model. But the second CCA step is essential for further improving the group robustness of the CLIP model.

### 4.4.3 Effect of the Data Augmentation

We further study the effect of data augmentation on the group robustness of the CLIP-ViT-L/14 on CelebA. Then, we consider four different scenarios: (a) only using the original prompt (S1); (b) using the original prompt and random character sequences (S2); (c) using the original prompt and its variants that contain the attribute information (S3); (d) using the original prompt and LLM-generated sentences (S4); (e) using the original prompt, random sentences, and LLM-generated sentences (S5); The results are shown in Figure 8. We can observe that different data augmentation methods affect performance. If we only use the original prompts, the results are worse. This verifies our assumption that the original prompt is not enough to learn stable transformation matrices. Moreover, although random sentences contribute to performance improvements, their average is the lowest. Thus, when we combine them with LLM-generated sentences, the performance can be further improved, achieving a good trade-off between average accuracy and group robustness. It's worth noting that we employ an LLM only once per class to generate a small set of auxiliary sentences that serve as a form of data augmentation. And we do not invoke the LLM during inference, so that the inference of our method is efficient on par with the original CLIP (see Section 4.5). Furthermore, different from previous work Yang et al. (2024), our approach does not require complex, meticulously designed processes. It only utilizes LLMs to assist in generating a small number of sentences, thereby enhancing textual diversity and stabilizing CCA optimization.

---

[4]The dimension of the HiT feature is 384.

Table 3: Performance comparison across proxy types.

| Proxy Type | Waterbirds Avg | Waterbirds Worst | CelebA Avg | CelebA Worst |
|---|---|---|---|---|
| Text Embedding (ours) | 89.55 | 62.50 | 85.35 | 83.00 |
| Real Image Features | 91.93 | 64.29 | 88.68 | 84.35 |

Table 4: The results of computational cost on Waterbirds and CelebA. We compare our method with the original CLIP, and we select two backbones: ResNet-50 and ViT-L/14.

| Model | Method | Waterbirds | | | | CelebA | | | |
|---|---|---|---|---|---|---|---|---|---|
| | | Time(s) | CPU(MB) | GPU(MB) | FLOPs(G) | Time(s) | CPU(MB) | GPU(MB) | FLOPs(G) |
| RN50 | Origin | 40.67 | 3,355 | 1,699 | 4.1 | 251.67 | 7,282 | 1,695 | 4.1 |
| | Ours | 41.10 | 3,359 | 1,733 | 4.51 | 252.82 | 7,315 | 1,733 | 4.51 |
| ViT-L/14 | Origin | 50.63 | 5,565 | 3,209 | 81.1 | 577.62 | 8,123 | 3,209 | 81.1 |
| | Ours | 51.13 | 5,622 | 3,243 | 89.21 | 579.27 | 8,165 | 3,243 | 89.21 |

#### 4.4.4  Effect of Using Text as an Image Proxy

In the second step, we use the original text embeddings $X$ as a proxy for image features. This is because we focus on zero-shot inference, so we do not access any training or validation images from the target dataset and cannot directly leverage image features from the target domain. Therefore, to validate this design, we conduct a new ablation study in a setting where labeled target-domain image data are available, replacing the class-level text embeddings with real image features. The results are reported in Table 3. We observe that using features extracted from real images indeed yields better performance, improving worst-group accuracy by 1–2% on average compared to the text-based proxy. This result empirically validates that class-level text embeddings serve as a reasonable proxy for target-domain image features under a zero-shot setting.

Then, we analyze the rationality of this design from a theoretical perspective. In our work, we focus on zero-shot inference; therefore, we do not access any training or validation images from the target dataset and cannot directly leverage image features from the target domain. To mitigate this limitation, we exploit CLIP's inherent text–visual alignment: for semantically aligned text–image pairs (T, I), the expected class-conditional text and image embeddings are approximately equal, i.e., $\mathbb{E}[\Phi_t(T)|y] \approx \mathbb{E}[\Phi_v(I)|y]$ as formalized in Proposition 1 (Appendix D). Consequently, under the constraint of no access to target-domain images, class-level text embeddings serve as our best available proxy for the image feature distribution of each class.

### 4.5  The Analysis of Efficiency

Finally, we evaluate the efficiency of DoubleCCA on the full test sets of Waterbirds and CelebA. We provide quantitative results, including total inference time, CPU/GPU memory consumption, and FLOPs. Here, we report the results for CLIP-RN50 and CLIP-ViT-L/14 in Table 4, and more results can be found in the Appendix H. First, we observe that the FLOPs increase by approximately 10% compared to the original CLIP model. Second, wall-clock inference time only increases by approximately 0.3–1.6%, and GPU/CPU memory increases by approximately 1–3%. These results show that DoubleCCA incurs negligible additional computational overhead during inference, with inference speed remaining virtually unchanged. This aligns with our complexity analysis and empirically validates the claim that DoubleCCA is computationally efficient.

## 5  Related Work

We briefly review related work on group robustness. Group robustness is a critical issue in machine learning, especially in the context of fairness and bias. Many works focus on improving the group robustness of foundation models. Existing methods can be divided into three categories: prompt tuning, adapter-based methods, and fine-tuning methods. The first category includes methods that modify the input prompts given to a pre-trained model to better align with the desired output. Representative works include (Chuang et al., 2023; Phan et al., 2024; Yang et al., 2024). The second category includes methods that add additional

modules to the pre-trained model to adapt it to the target task. Representative works include (Zhang & Ré, 2022; Gao et al., 2024; Dehdashtian et al., 2024c). The third category includes methods that fine-tune the pre-trained model on the target task. The representative works include (Kumar et al., 2022). In addition to these methods, An et al.(An et al., 2024) proposes a perception-aware method (called PerceptionCLIP) to enhance the group robustness of the CLIP model, which provides CLIP with contextual attributes. This is similar to our method, which also enriches the text embeddings of the CLIP model with additional semantic information. Both methods aim to improve the group robustness of the zero-shot classifier. According to the experiments, our method outperforms PerceptionCLIP in terms of both average accuracy and worst-group robustness. Since our method is simple and easy to implement, it can be easily integrated into existing models, such as the contrastive adapter (Zhang & Ré, 2022), providing a practical solution to improve the robustness of the foundation models.

Finally, we clarify how our approach differs from prior CCA-based methods (a summary can refer to Table 10 in the Appendix). The first CCA step in our framework resembles classical multi-view CCA approaches Dhillon et al. (2011), which align CLIP text embeddings with auxiliary sentence-level representations in a shared low-rank subspace. However, unlike these works, we introduce a second CCA step that explicitly projects the fused embedding back into the original CLIP vision-compatible feature space. This design ensures compatibility with frozen CLIP inference while preserving semantic richness. In contrast to Horoi et al. Horoi et al. (2024), which requires extracting and fusing multi-layer internal features from trained vision-language models (and often involves fine-tuning or architectural modifications), our method operates entirely on class-level text embeddings using closed-form CCA transformations. Crucially, it does not require access to training images, modifies neither the vision nor the language backbone, and avoids any fine-tuning. As a result, our approach is a lightweight, plug-and-play module that can be seamlessly integrated with existing group-robustness frameworks such as PerceptionCLIP and Contrastive Adapter.

## 6 Conclusion

We proposed DoubleCCA, a novel method to improve the robustness of foundation models to group-based biases. By employing CCA twice, our method effectively aligns and merges different text representations. We demonstrated the effectiveness of DoubleCCA on various datasets, showing that it outperforms existing methods in terms of both group robustness and domain generalization. Our approach is simple to implement and can be easily integrated into existing models, providing a practical solution to improve the robustness of foundation models. Future work could explore the theoretical foundations of this approach and further design a black-box optimization scheme (Song et al., 2024) to enhance robustness. On the other hand, we also plan to explore the effectiveness of our approach in more complex, non-object-centric settings, such as scene understanding, multi-label classification, or context-dependent recognition tasks.

### Acknowledgments

This work is supported by the National Science Fund for Distinguished Young Scholars (No.62025603) and the China Postdoctoral Science Foundation (No.2025M771514).

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

# A    Visualization results on CelebA

Figure 9 shows similar patterns on the CelebA dataset, where arrows connect same-class centroids across gender attributes (Male → Female). CLIP's representations exhibit large gender-induced shifts, as indicated by long arrows. In contrast, auxiliary encoders keep within-class clusters compact and are less sensitive to spurious gender attributes. Complementing these visualizations, Figure 10 provides a quantitative summary of the centroid shifts by reporting the mean $\ell_2$ distance between male and female centroids within each class (lower is better) and confirms that auxiliary encoders consistently yield smaller cross-gender distances than CLIP, consistent with the t-SNE trends.

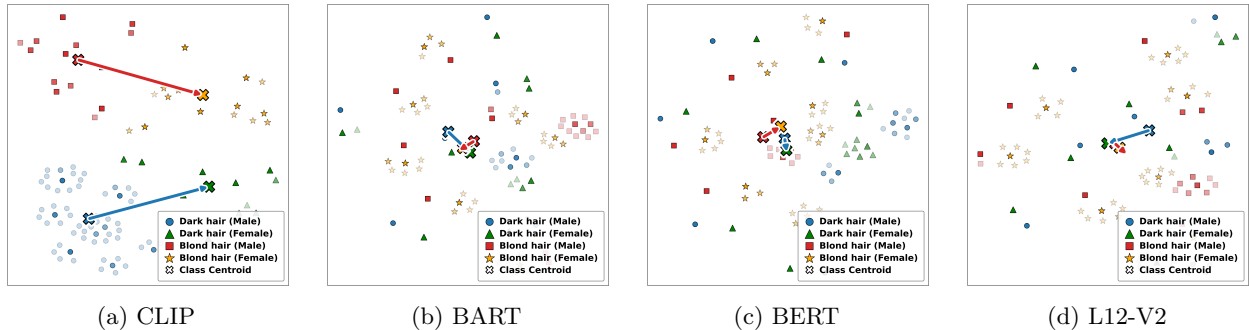

(a) CLIP      (b) BART      (c) BERT      (d) L12-V2

Figure 9: Group-bias visualization of text embeddings across text encoders on the CelebA dataset. Each panel shows 2D t-SNE projections from: (a) CLIP, (b) BART, (c) BERT, (d) L12-V2. Points represent text embeddings colored by class-gender combinations. Arrows connect same-class centroids across gender attributes (Male → Female). The substantially shorter arrows in BART, BERT, and L12-V2 compared to CLIP indicate these auxiliary encoders learn representations that are more robust to spurious gender correlations.

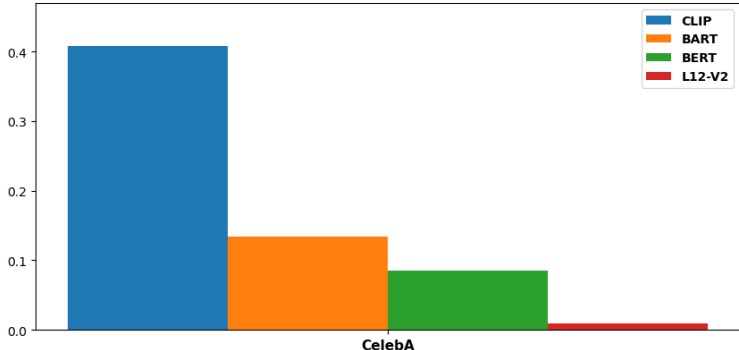

Figure 10: Quantitative comparison of attribute bias across text encoders on CelebA datasets. Attribute bias measures the average L2 distance between class-conditional attribute centroids in the embedding space (see Appendix B for details). Lower values indicate more invariant representations.

# B    Attribute Bias Score Computation

The Attribute Bias Score (ABS) used in Figure 3 and Figure 10 quantifies how much text representations shift due to spurious attributes within each class. Given a text encoder $\Phi$, we compute:

- **Class-attribute centroids**: For each class $c \in \mathcal{C}$ and attribute $a \in \mathcal{A}$, we compute the centroid of all text embeddings:

$$\boldsymbol{\mu}_{c,a} = \frac{1}{N_{c,a}} \sum_{i:y_i=c,a_i=a} \Phi(\mathbf{t}_i) \tag{12}$$

  where $N_{c,a}$ is the number of text samples with class $c$ and attribute $a$, and $\mathbf{t}_i$ represents the $i$-th text prompt.

- **Within-class attribute shift**: For each class $c$, we measure the $L_2$ distance between centroids across different attributes:

$$d_c = \|\boldsymbol{\mu}_{c,0} - \boldsymbol{\mu}_{c,1}\|_2 \tag{13}$$

- **Overall Attribute Bias Score**:

$$\text{ABS}(\Phi) = \frac{1}{|\mathcal{C}|} \sum_{c \in \mathcal{C}} d_c \tag{14}$$

  where $|\mathcal{C}|$ is the number of classes.

For the **Waterbirds** dataset, this metric captures how text representations of "landbird" and "waterbird" shift between land and water backgrounds. For **CelebA**, it measures shifts between male and female attributes for hair color classes. A lower ABS indicates that the encoder produces more invariant representations that are robust to spurious correlations.

## C   Integration with Existing Methods

DoubleCCA can be easily integrated with other robustness methods:

- **With PerceptionCLIP:** Replace their text embeddings with our merged embeddings $W$

- **With Contrastive Adapters:** Use $W$ as input to the adapter module

- **With prompt engineering methods:** Apply DoubleCCA to any engineered prompts

The modular nature of our approach allows it to enhance existing methods without architectural changes.

## D   Theoretical Motivation

**Semantic Structure.** Let $\mathcal{T}$ be the space of text descriptions. The semantic structure is characterized by a similarity function

$$s : \mathcal{T} \times \mathcal{T} \longrightarrow [0,1]$$

that captures linguistic relationships between texts.

**Structure Preservation.** An encoder $\Phi : \mathcal{T} \to \mathcal{E}$ preserves semantic structure if, for some small $\epsilon > 0$, it holds that

$$\forall\, t_1, t_2 \in \mathcal{T} : \quad \left| s(t_1, t_2) \,-\, \cos\big(\Phi(t_1), \Phi(t_2)\big) \right| < \epsilon.$$

**Proposition 1.** The CLIP text encoder $\Phi_t : \mathcal{T} \to \mathcal{E}_{\text{CLIP}}$, being optimized for vision–language alignment, induces a bias towards visual discriminability that can distort purely linguistic semantic structure—particularly when classes exhibit spurious visual correlations.

*Proof.* Consider the CLIP training objective

$$\mathcal{L}_{\text{CLIP}} = -\mathbb{E}_{(I,T) \sim \mathcal{D}} \left[ \log \frac{\exp\langle \Phi_v(I), \Phi_t(T) \rangle / \tau}{\sum_{T' \in \mathcal{B}} \exp\langle \Phi_v(I), \Phi_t(T') \rangle / \tau} \right].$$

Taking the gradient with respect to the text embedding yields a term that pushes $\Phi_t(T)$ toward its paired visual feature $\Phi_v(I)$ and away from others. If two semantically similar texts $t_1, t_2$ (e.g. "waterbird" vs. "seabird") happen to co-occur with very different visual contexts $I_1, I_2$, then to satisfy

$$\langle \Phi_t(t_1), \Phi_v(I_1) \rangle \gg \langle \Phi_t(t_1), \Phi_v(I_2) \rangle \quad \text{and} \quad \langle \Phi_t(t_2), \Phi_v(I_2) \rangle \gg \langle \Phi_t(t_2), \Phi_v(I_1) \rangle,$$

one must force $\|\Phi_t(t_1) - \Phi_t(t_2)\|$ large, contradicting their high linguistic similarity. Hence, the encoder trades off semantic preservation for visual discriminability when spurious cues are strong.

**Corollary 1.** For datasets with strong spurious correlations (strength $\rho$) between visual features and class labels, the expected distortion in the text-embedding space grows proportionally to $\rho$.

*Proof Sketch.* One can show

$$\mathbb{E}\big[\|\Phi_t(t_1) - \Phi_t(t_2)\|\big] \gtrsim c\,\rho\,\big\|\mathbb{E}[\Phi_v(I_1)] - \mathbb{E}[\Phi_v(I_2)]\big\|,$$

for some constant $c > 0$ depending on $\tau$ and $|\mathcal{D}|$.

**Lemma (Information Decomposition).** For any text $T \in \mathcal{T}$, its total information $H(T)$ decomposes as

$$H(T) = I(T; \mathcal{V}) + I\big(T; \mathcal{L} \mid \mathcal{V}\big) + H\big(T \mid \mathcal{V}, \mathcal{L}\big),$$

where $\mathcal{V}$ denotes visual concepts and $\mathcal{L}$ denotes pure linguistic structure.

**Proposition (Complementary Encoders).** Let $\Phi_t$ be CLIP's text encoder and $\Phi_{se}$ a sentence-only encoder. Then

$$I\big(\Phi_t(T); \mathcal{V}\big) > I\big(\Phi_{se}(T); \mathcal{V}\big), \quad I\big(\Phi_{se}(T); \mathcal{L} \mid \mathcal{V}\big) > I\big(\Phi_t(T); \mathcal{L} \mid \mathcal{V}\big).$$

*Proof Sketch.* CLIP's objective maximizes alignment with visual features (hence upper-bounds $I(\Phi_t(T); \mathcal{V})$), whereas sentence encoders optimize for textual mutual information across related sentences (preserving linguistic structure).

# E   Additional Ablation Studies on Data Augmentation

To further investigate the contribution of each component in our data augmentation strategy, we conducted detailed ablation experiments on the CelebA dataset using ViT-L/14. We systematically evaluated four different configurations:

- **S-(1)**: Baseline using only the original CLIP prompt template "a photo of a {class}"

- **S-(2)**: Original prompt augmented with random character sequences

- **S-(3)**: Original prompt augmented with LLM-generated contextual descriptions

- **S-(4)**: Full augmentation combining original prompt, random sequences, and LLM-generated descriptions

Table 5: Ablation study of data augmentation strategies on CelebA (ViT-L/14). S-(1): original prompt only; S-(2): original + random sequences; S-(3): original + LLM descriptions; S-(4): all combined.

|       | S - (1) | S - (2) | S - (3) | S - (4) |
|-------|---------|---------|---------|---------|
| Worst | 51.44   | 38.93   | 67.73   | 88.24   |
| Avg   | 52.84   | 39.47   | 68.82   | 88.35   |

The results in Table 5 reveal several key insights:

**Random sequences alone are detrimental** (S-2): Using only random character augmentation significantly degrades performance (38.93% worst-group accuracy vs. 51.44% baseline), confirming that meaningful semantic augmentation is crucial.

**LLM-generated descriptions provide improvements** (S-3): Augmenting with semantically meaningful LLM-generated sentences improves the worst-group accuracy to 67.73%, demonstrating the value of coherent linguistic variations.

**Combining all augmentation strategies is optimal** (S-4): The full augmentation strategy achieves the best performance (88.24% worst-group accuracy), suggesting that the diversity from both random perturbations and semantic variations helps CCA learn more robust transformation matrices.

These findings validate our design choice of using combined augmentation (S-4) in the main experiments, as it provides sufficient data diversity for stable CCA estimation while maintaining semantic coherence.

## F  Details of Inference Protocol

In this supplementary section, we provide a detailed description of the inference protocol employed in Tables 1 and 2. Such clarification is crucial for understanding the evaluation pipeline and for enabling reliable reproduction of our findings.

### F.1  Inference Protocol Based on PerceptionCLIP

In fact, we strictly adhered to PerceptionCLIP's experimental setup to ensure a fair comparison. In Tables 1–2, we adopt its two-stage inference pipeline with attribute combinations such as "+background," "+gender," "+gender, age," and "+gender, age, race," without incorporating any additional information beyond what is specified in PerceptionCLIP. Here, we first introduce the details of the two-stage inference pipeline.

**Stage 1: Attribute Inference**. Given a test image, we follow PerceptionCLIP's implementation: CLIP computes similarity scores between the image and a set of predefined attribute templates—e.g., "on grass," "in the water," "in a forest," "with sky," "on a beach," … for Waterbirds, and "male"/"female," "young"/"old," "light skin"/"dark skin"/"Asian"/"other" for CelebA. These scores yield an approximate posterior distribution $p(z \mid I)$, where *z* denotes contextual attributes such as background, gender, age, or race. Critically, all such attributes are predicted by the model at inference time and do not rely on any ground-truth annotations from the dataset.

**Stage 2: Conditional Classification and Aggregation** For each candidate contextual attribute *z* (e.g., background for Waterbirds; gender and age for CelebA), we construct class-conditional prompts that incorporate context:

- **Waterbirds**: "a photo of a {class} on {background}"
- **CelebA**: "a photo of a {gender} {age} person with {hair color}"

We then use CLIP (or CLIP + DoubleCCA) to compute the conditional class probability $p(y \mid I, z)$. Following PerceptionCLIP, we perform a weighted aggregation to approximate marginalization over z, which uses the estimated posterior $p(z \mid I)$ to obtain the final prediction $p(y \mid I) = \sum_z p(y \mid I, z)p(z \mid I)$. Crucially, this entire process is unsupervised at inference time: no ground-truth attributes are used—only model-generated estimates. As noted in Section 4.2 of the original manuscript: "We only replace the original prompt embedding with the merged text embedding W and adopt the same inference process as PerceptionCLIP."

### F.2  Inference Protocols Across Experimental Settings

First, we provide a row-by-row breakdown of the inference protocol for the main entries in Tables 1–2, explicitly indicating whether ground-truth group labels are used. In all cases, no ground-truth group information is used at any stage of inference. The details are described in Table 6.

Second, we present the details of the inference protocol for the Waterbirds dataset.

Table 6: Inference protocol for each row in Tables 1, 2. All methods use predicted attributes at test time; ground-truth group labels are never used in prompts or model inputs.

| Setting | Inference Protocol Summary | Uses GT Group Label? |
|---|---|---|
| CLIP | Standard zero-shot: uses template "a photo of a {class}", predicts $\hat{y} = \arg\max_y \cos(\Phi_t(T_y), \Phi_v(I))$. | No |
| Ours (DoubleCCA) | Same zero-shot inference pipeline as CLIP, except all $\Phi_t(T_y)$ are replaced with our merged embeddings $W_y$: $\hat{y} = \arg\max_y \cos(W_y, \Phi_v(I))$. | No |
| +background +gender +gender,age +gender,age,race (PerceptionCLIP) | Follows exactly the two-stage pipeline of PerceptionCLIP: (1) First uses CLIP to predict the context distribution $p(z \mid I)$; (2) Then performs conditional classification under each candidate context and aggregates. | No |
| Ours + background Ours + gender Ours + gender,age Ours + gender,age,race | Based on the above PerceptionCLIP pipeline, only replacing all text embeddings $\Phi_t(\cdot)$ with our pre-computed merged embeddings $W$; attributes are still automatically predicted and marginalized by the model. | No |

- **CLIP** and **Ours**: Use only class-name prompts (e.g., "a photo of a {class}") without incorporating any background information.

- **+background**: The background attribute is first predicted by CLIP using a set of background-specific attribute templates (e.g., "on grass," "in the water"). The top predicted background is then injected into the class prompt (e.g., "a photo of a [class] on grass") to perform conditional classification. This two-stage inference pipeline exactly follows the original PerceptionCLIP protocol.

- **Ours + background**: We replace all text embeddings in the above PerceptionCLIP pipeline with the merged embedding W produced by DoubleCCA. All other steps—including background prediction and prompt composition—remain unchanged.

Third, we present the details of the inference protocol for the CelebA dataset.

- **CLIP** and **Ours**: Use only hair-color class-name prompts (e.g., "a photo of a person with blond hair"), without incorporating gender, age, or race.

- **+gender/+gender,age/+gender,age,race**: These settings treat gender, gender+age, and gender+age+race as contextual factors z. At test time, all such attributes are predicted by the model from the input image using CLIP-based attribute templates and dynamically inserted into the prompt (e.g., "a photo of a young female person with blond hair"). No ground-truth attributes are used during inference; they are employed only during evaluation to define groups for computing worst-group and average metrics.

- **Ours + . . .**: For each of the above configurations, we adopt the same attribute vocabulary, prompt templates, and inference protocol as the corresponding PerceptionCLIP variant. The only difference lies in the text representation: we replace CLIP's original text embeddings with the merged embedding W produced by DoubleCCA.

Consequently, across all rows involving contextual attributes, our method maintains alignment with PerceptionCLIP's inference pipeline. The sole modification is the underlying text embedding—ensuring that performance differences stem solely from representational quality, not protocol discrepancies. This guarantees a fair and controlled comparison.

# G   Experiments on Out-of-Distribution and Complex Real-world Scenarios

## G.1   Experimental Setting

To further validate the effectiveness of our method, we evaluate it on five diverse robustness benchmarks:

- **CMNIST** (LeCun et al., 2002): A synthetic variant of MNIST with controlled color–digit spurious correlations.

- **FairFace** (Karkkainen & Joo, 2021): A real-world facial dataset widely used for evaluating fairness across age, gender, and race attributes.

- **CounterAnimal** (Wang et al., 2024): A benchmark featuring animal classification under spurious correlations and domain shifts.

- **ImageNet-A** (Hendrycks et al., 2021): A challenging out-of-distribution (OOD) benchmark composed of natural adversarial examples that fool standard models.

- **COCO-FP** (Liu et al., 2024): A multi-object scene dataset derived from COCO, designed to test robustness in the presence of distractor objects and complex backgrounds.

On all five datasets, we compare our approach against the original CLIP using the same five robustness metrics, whose details are shown as follows:

- **Average Acc**: Mean accuracy across all samples.

- **Worst Acc**: Accuracy of the worst-performing spurious attribute group.

- **Class-wise Robust**: For each class, compute the worst-group accuracy among its subgroups; then average these values across all classes.

- **Worst-10 Groups**: Mean accuracy of the 10 worst-performing groups (by accuracy).

- **Robust@95%**: The 95th percentile robust accuracy—obtained by sorting all groups by accuracy, discarding the bottom 5%, and reporting the minimum accuracy among the remaining 95%.

## G.2   Experimental Results

The results are reported in Table 7. On FairFace and CounterAnimal, DoubleCCA significantly improves robustness metrics such as worst-group/Robust@95% while maintaining similar or slightly improved average accuracy (e.g., Robust@95% improves from 50.00% to 57.14% on CounterAnimal; from 42.60% to 47.00% on FairFace); On datasets such as CMNIST, DoubleCCA improves multiple robustness metrics, with small average accuracy trade-offs in some scenarios, consistent with the fairness-accuracy trade-off observed on Waterbirds/CelebA; On the more challenging ImageNet-A/COCO-FP, although the worst-group accuracy is limited by dataset characteristics, DoubleCCA shows slight improvements or maintains parity on more stable metrics such as Class-wise Robust (e.g., Class-wise Robust improves from 65.92% to 67.36% on COCO-FP), at least not worse than the original CLIP. These results demonstrate that the benefits of DoubleCCA are not limited to Waterbirds/CelebA, but can generalize to more realistic and challenging tasks.

Moreover, we explain why the worst group accuracy is 0% on ImageNet-A and COCO-FP. On both datasets, both DoubleCCA and the original CLIP achieve 0% worst-group accuracy. This is not indicative of a methodological failure, but rather stems from the inherent characteristics of these benchmarks—notably, extreme label imbalance and the presence of groups with very few (or zero) test samples from certain classes. Consequently, we caution against interpreting worst-group accuracy in isolation on such highly imbalanced and long-tailed benchmarks.

Therefore, given the inherent difficulty of benchmarks like ImageNet-A and COCO-FP, relying solely on worst-group accuracy yields unstable and overly evaluations: performance is dominated by a handful of groups with

Table 7: Results of ViT-L/14 backbone on five different benchmark datasets.

| Dataset | Method | Average Acc | Worst Acc | Class-wise Robust | Worst-10 Groups | Robust@95% |
|---|---|---|---|---|---|---|
| CMNIST | Origin | 73.76 | 44.74 | 64.47 | 58.15 | 52.01 |
| | Ours | 71.15 | 46.78 | 67.22 | 60.33 | 52.81 |
| Fairface | Origin | 67.63 | 41.00 | 65.93 | 62.57 | 42.60 |
| | Ours | 68.76 | 43.50 | 66.84 | 63.10 | 47.00 |
| CounterAnimal | Origin | 90.00 | 10.26 | 86.30 | 43.08 | 50.00 |
| | Ours | 90.61 | 12.82 | 86.69 | 47.46 | 57.14 |
| Imagenet-A | Origin | 67.68 | 0.00 | 55.98 | 7.43 | 20.00 |
| | Ours | 67.07 | 0.00 | 56.13 | 7.43 | 20.00 |
| CoCo-FP | Origin | 52.11 | 0.00 | 65.92 | 7.16 | 8.33 |
| | Ours | 50.64 | 0.00 | 67.36 | 5.84 | 8.33 |

only a few samples, where outcomes are highly sensitive to statistical noise. To enable a more comprehensive and stable assessment of group robustness, we complement worst-group accuracy with three additional metrics: Class-wise Robust Accuracy, Worst-10 Groups Average, and Robust@95%. These evaluation metrics can provide complementary perspectives that better reflect model behavior under long-tailed group distributions.

## H    Additional Results of Efficiency

Table 8: Computational cost on Waterbirds and CelebA.

| Model | Method | Waterbirds | | | | CelebA | | | |
|---|---|---|---|---|---|---|---|---|---|
| | | Time(s) | CPU(MB) | GPU(MB) | FLOPs(G) | Time(s) | CPU(MB) | GPU(MB) | FLOPs(G) |
| RN50 | Origin | 40.67 | 3,355 | 1,699 | 4.1 | 251.67 | 7,282 | 1,695 | 4.1 |
| | Ours | 41.10 | 3,359 | 1,733 | 4.51 | 252.82 | 7,315 | 1,733 | 4.51 |
| ViT-B/32 | Origin | 42.01 | 3,650 | 1,135 | 4.4 | 248.78 | 5,626 | 1,135 | 4.4 |
| | Ours | 42.69 | 3,712 | 1,173 | 4.84 | 250.25 | 5,635 | 1,173 | 4.84 |
| ViT-B/16 | Origin | 51.97 | 3,589 | 1,865 | 17.6 | 246.49 | 5,672 | 1,865 | 17.6 |
| | Ours | 52.80 | 3,594 | 1,903 | 19.36 | 248.05 | 5,696 | 1,903 | 19.36 |
| ViT-L/14 | Origin | 50.63 | 5,565 | 3,209 | 81.1 | 577.62 | 8,123 | 3,209 | 81.1 |
| | Ours | 51.13 | 5,622 | 3,243 | 89.21 | 579.27 | 8,165 | 3,243 | 89.21 |

To further validate the effectiveness of our proposed method, we report supplementary results on two additional backbone models in the appendix. Specifically, we evaluate our approach on CLIP ViT-B/16 and CLIP ViT-B/32 using the same experimental settings and evaluation metrics as in the main text. The results are summarized in Table 8. Consistent with the findings in the main paper, our method achieves improved group robustness and maintains competitive average accuracy across both additional models. These supplementary results confirm that our proposed DoubleCCA framework consistently enhances model robustness to spurious correlations, regardless of the choice of backbone.

## I    Clarification on the Use of Race Attributes in CelebA

Since the CelebA dataset does not include race annotations by default, we clarify the origin and usage of these attributes in the appendix.

First, we would like to clarify that the race attributes mentioned in our manuscript, such as "White", "Black", and "Asian", are neither manual annotations we produced nor labels generated by an external classifier. It should be noted that the race attributes are used here for the purpose of conducting a strict and fair comparison with PerceptionCLIP An et al. (2024), and thus we fully adhere to the list of contextual attributes defined in the open-source code of this paper (as shown in Table 9).

Table 9: Contextual attributes and their descriptions for Waterbirds and CelebA datasets, adopted from PerceptionCLIP An et al. (2024).

| Dataset | Domain Template | Attributes | Values |
|---|---|---|---|
| Waterbirds | "a photo of a {y}" | *background*
*background*$^+$ | on land, on water
+ in forest, in sky, on street, on grass, on tree,
    with flowers, on beach, with human, on a branch |
| CelebA | "a photo of a celebrity with {y}" | *gender*
*age*
*race* | female, male
young, old
white skin, dark skin, asian |

Second, in our experimental setup (corresponding to the "+race" rows in Table 1), the race-related terms (e.g., "White", "Black", and "Asian") are used exclusively as prompt candidates during text embedding construction and do not serve as ground-truth labels for evaluation. Specifically, the model computes the similarity between the input image and a set of attribute-specific text descriptions during inference. It then dynamically infers the most likely attribute (e.g., race or gender) by treating the CLIP model itself as a soft classifier. This inferred attribute is subsequently injected as contextual information into the text encoder to aid the primary classification task (e.g., hair color prediction).

Third, it's worth noting that the "Worst-group Accuracy" reported in Table 1 is still computed based on the attributes officially provided by CelebA, not based on race. Race information is used only as auxiliary input context and does not participate in evaluation grouping.

Therefore, our method does not require CelebA to include race labels, nor have we performed any additional race annotation on the dataset. We will explicitly clarify this point in the revised manuscript.

Table 10: Comparison of DoubleCCA with existing CCA-based methods.

| Method | Primary Goal | Computational Characteristics | Typical Application Scenario |
|---|---|---|---|
| Traditional CCA | Feature alignment | Closed-form, efficient | Multi-view representation learning, dimensionality reduction |
| Horoi et al., 2024 | Model fusion | Requires access to multi-layer features | Merging trained networks with identical architectures |
| DoubleCCA (ours) | Debiasing / improving group robustness | Closed-form, negligible inference overhead | CLIP zero-shot classification + text-side debiasing |

