# OpenReview forum: "Improving Foundation Model Group Robustness with Auxiliary Sentence Embeddings"
_TMLR — Accepted by TMLR_

### Review · Reviewer_oKVQ · 2025-10-09

**Summary Of Contributions:**

The paper proposes DoubleCCA, a two-stage Canonical Correlation Analysis pipeline that enriches CLIP’s text side using an auxiliary sentence-embedding model (e.g., HiT, SBERT, GTE). Stage 1 aligns CLIP text embeddings with auxiliary sentence embeddings in a shared space; Stage 2 merges these predictors and projects back to CLIP’s text space to maintain compatibility with image features. To stabilize CCA when there are few classes, the authors augment prompts with random strings and LLM-generated sentences. Experiments on Waterbirds and CelebA report gains in worst-group accuracy and smaller robustness gaps; DoubleCCA is also shown to combine with methods like PerceptionCLIP, Oth-Cal, and Contrastive Adapters. The method is plug-and-play, closed-form (eigendecomposition), and claims negligible overhead under typical settings.

**Audience:**

Yes

**Audience Explanation:**

Since this work is about improving CLIP classification by embedding more semantic context, there will be TMLR's audience interested in knowing the findings of this paper.


## Weaknesses

1. Can this method use extra sentences to describe very complex scenes and non-object-centric scenarios?
This approach augments class-level prompts, but is not evaluated on multi-entity/relationship-centric tasks. Extending to long captions or region-level prompts may require architectural changes (e.g., region-level text embeddings, grounding), which this paper did not discuss.

2. Extra computation costs to embed more sentences; any study comparing costs vs improvements?
In this paper, only asymptotic complexity and a qualitative “negligible” claim are given; no runtime, memory, or cost–benefit curves versus alternatives.


3. Can this method scale up in real cases? test-time scaling law (improvements vs extra embedding cost)?
The paper provides sensitivity (K, dim) but no explicit scaling law plotting robustness gains against incremental test-time compute/latency.



4. Main test cases are too small and not representative.
Evaluation is confined to Waterbirds/CelebA. Broader stress tests are needed.

**Broader Impact Concerns:**

This work does not introduce any ethical concerns.

**Claims And Evidence:**

Yes

**Claims Explanation:**

## Strengths

1. Simple, modular plug-in: Works with frozen CLIP; easy to combine with existing debiasing methods (PCLIP, Oth-Cal, adapters).

2. Clear intuition + concrete procedure: Uses auxiliary text encoders to counter text-side spurious shifts; well-specified two-CCA pipeline with algorithm summary.

3. Comprehensive ablations: Number of augmented sentences, CCA dimension, effect of each CCA stage, and alternative sentence encoders (HiT/GTE/BERT/BART).

**Requested Changes:**

1. Benchmark breadth & realism.
Add evaluations of real-world or large-scale benchmarks, such as ImageNet, FairFace/UTKFace, and at least one multi-object dataset (e.g., COCO zero-shot classification with distractor backgrounds).

2. Include an OOD shift benchmark (domain shifts or spurious attributes beyond those seen in Waterbirds/CelebA).

3. Discuss and study computation costs and scaling law.
Report text-embedding latency, FLOPs, GPU/CPU memory for embedding generation and CCA at test time; compare to strong baselines (prompt ensembling, adapters, minimal fine-tuning).

4. Complex scenes & non-object-centric settings.
Evaluate on scene-centric prompts (Places, COCO-style captions) or add a region-text variant (e.g., apply DoubleCCA to region prompts from a grounding model), and measure robustness under context confounders.

---

> ### Author Response · Authors · 2025-12-10
> **Official Comment by Authors [1/4]**
>
> **New Benchmark Evaluation: OOD and Complex Scenarios**
>
> We thank the reviewer for this insightful suggestion. In response, we have conducted a series of new experiments on out-of-distribution (OOD) benchmarks and more complex real-world scenarios. We compare our method with original CLIP on five new benchmark dataset, including CMNIST, FairFace, CounterAnimal, ImageNet-A and COCO-FP.
>
> The summarization of these new added dataset are shown as follows:
>
> - **CMNIST**: Synthetic colored MNIST with controlled spurious correlations;
> - **FairFace**: A real-world facial attribute dataset for face fairness evaluation;
> - **CounterAnimal**: A dataset containing animal-related spurious correlations and domain shifts;
> - **ImageNet-A**: A highly challenging real-world OOD classification benchmark;
> - **COCO-FP**: Multi-object / distractor background scenarios based on COCO.
>
> On these datasets, we report the following robustness metrics:
>
> - **Average Accuracy**: Mean accuracy across all samples.
> - **Worst-Group Accuracy (Worst Acc)**: Accuracy of the worst-performing spurious attribute group.
> - **Class-wise Robust Accuracy**: For each class, compute the worst-group accuracy among its subgroups; then average these values across all classes.
> - **Worst-10 Groups Average**: Mean accuracy of the 10 worst-performing groups (by accuracy).
> - **Robust@95%**: The 95th percentile robust accuracy—obtained by sorting all groups by accuracy, discarding the bottom 5%, and reporting the minimum accuracy among the remaining 95%.
>
> The results are shown in the following Table:
>
> | Dataset         | Method | Average Acc | Worst Acc | Class-wise Robust | Worst-10 Groups | Robust@95% |
> |-----------------|--------|-------------|-----------|-------------------|------------------|------------|
> | CMNIST          | Origin | 73.76%      | 44.74%    | 64.47%            | 58.15%           | 52.01%     |
> |                 | Ours   | 71.15%      | 46.78%    | 67.22%            | 60.33%           | 52.81%     |
> | Fairface        | Origin | 67.63%      | 41.00%    | 65.93%            | 62.57%           | 42.60%     |
> |                 | Ours   | 68.76%      | 43.50%    | 66.84%            | 63.10%           | 47.00%     |
> | CounterAnimal   | Origin | 90.00%      | 10.26%    | 86.30%            | 43.08%           | 50.00%     |
> |                 | Ours   | 90.61%      | 12.82%    | 86.69%            | 47.46%           | 57.14%     |
> | Imagenet-A      | Origin | 67.68%      | 0.00%     | 55.98%            | 7.43%            | 20.00%     |
> |                 | Ours   | 67.07%      | 0.00%     | 56.13%            | 7.43%            | 20.00%     |
> | CoCo-FP         | Origin | 52.11%      | 0.00%     | 65.92%            | 7.16%            | 8.33%      |
> |                 | Ours   | 50.64%      | 0.00%     | 67.36%            | 5.84%            | 8.33%      |
>
> **Experimental Results**
>
> On FairFace and CounterAnimal, DoubleCCA significantly improves robustness metrics such as worst-group / Robust@95% while maintaining similar or slightly improved average accuracy (e.g., Robust@95% improves from 50.00% to 57.14% on CounterAnimal; from 42.60% to 47.00% on FairFace);
>
> On datasets such as CMNIST, DoubleCCA improves multiple robustness metrics, with small average accuracy trade-offs in some scenarios, consistent with the fairness-accuracy trade-off observed on Waterbirds / CelebA;
>
> On the more challenging ImageNet-A / COCO-FP, although worst-group accuracy is limited by dataset characteristics, DoubleCCA shows slight improvements or maintains parity on more stable metrics such as Class-wise Robust (e.g., Class-wise Robust improves from 65.92% to 67.36% on COCO-FP), at least not worse than the original CLIP.
>
> These results demonstrate that the benefits of DoubleCCA are not limited to Waterbirds / CelebA, but can generalize to more realistic and challenging OOD and multi-scenario tasks.

---

> ### Author Response · Authors · 2025-12-10
> **Official Comment by Authors [2/4]**
>
> **Experimental Results**
>
> On FairFace and CounterAnimal, DoubleCCA significantly improves robustness metrics such as worst-group / Robust@95% while maintaining similar or slightly improved average accuracy (e.g., Robust@95% improves from 50.00% to 57.14% on CounterAnimal; from 42.60% to 47.00% on FairFace);
>
> On datasets such as CMNIST, DoubleCCA improves multiple robustness metrics, with small average accuracy trade-offs in some scenarios, consistent with the fairness-accuracy trade-off observed on Waterbirds / CelebA;
>
> On the more challenging ImageNet-A / COCO-FP, although worst-group accuracy is limited by dataset characteristics, DoubleCCA shows slight improvements or maintains parity on more stable metrics such as Class-wise Robust (e.g., Class-wise Robust improves from 65.92% to 67.36% on COCO-FP), at least not worse than the original CLIP.
>
> These results demonstrate that the benefits of DoubleCCA are not limited to Waterbirds / CelebA, but can generalize to more realistic and challenging OOD and multi-scenario tasks.
>
> Moreover, we clarify why the worst-group accuracy is 0% on ImageNet-A and COCO-FP. On both datasets, both DoubleCCA and the original CLIP achieve 0% worst-group accuracy. This is not indicative of a methodological failure, but rather stems from the inherent characteristics of these benchmarks—notably, extreme label imbalance and the presence of groups with very few (or zero) test samples from certain classes. Consequently, we caution against interpreting worst-group accuracy in isolation on such highly imbalanced and long-tailed benchmarks.
>
> Specifically, the 0% worst-group accuracy observed on ImageNet-A and COCO-FP arises from the fundamental properties of these benchmarks, not from the limitations of our method.
> We highlight three key factors:
>
> - **Extreme sample scarcity**: Unlike Waterbirds or CelebA—which feature carefully balanced group structures—ImageNet-A and COCO-FP contain numerous class–attribute combinations with only 5–6 (or even fewer) test samples. Under such extreme sparsity, it is statistically likely for a group’s accuracy to drop to 0%, a well-documented limitation of worst-group evaluation in highly imbalanced settings.
> - **Inherently high dataset difficulty**: ImageNet-A consists of “natural adversarial examples” specifically selected because they fool state-of-the-art models; COCO-FP emphasizes complex scenes with multiple objects and cluttered backgrounds. Many minority groups in these datasets lie near decision boundaries or contain highly ambiguous instances that even strong baselines like CLIP struggle to classify correctly.
> - **Baseline performance confirms dataset-driven behavior**: Our new results show that the original CLIP also achieves 0% worst-group accuracy on these benchmarks, confirming that the phenomenon is dataset-driven rather than caused by DoubleCCA degradation.
>
> Therefore, given these limitations of worst-group accuracy, we now justify the use of multiple complementary robustness metrics to provide a more reliable and nuanced assessment of model behavior.
>
> Given the inherent difficulty of benchmarks like ImageNet-A and COCO-FP, relying solely on worst-group accuracy yields unstable and overly pessimistic evaluations: performance is dominated by a handful of groups with only a few samples, where outcomes are highly sensitive to statistical noise. To enable a more comprehensive and stable assessment of group robustness, we complement worst-group accuracy with three additional metrics: **Class-wise Robust Accuracy**, **Worst-10 Groups Average**, and **Robust@95%**. These are not selected to “cherry-pick” favorable results, but rather to provide **complementary perspectives** that better reflect model behavior under long-tailed group distributions.
>
> This multi-metric evaluation aligns with established practices in the fairness and robustness literature, where single worst-case metrics are often supplemented with relaxed or averaged measures (e.g., conditional accuracy gaps, quantile-based robustness, or smoothed group risk) when evaluating models on highly imbalanced or sparse subgroup structures.
>
> We have added these new results and analysis in our revsion.

---

> ### Author Response · Authors · 2025-12-10
> **Official Comment by Authors [3/4]**
>
> **Response to Reviewer oKVQ's Point 4:**
>
> Our work focuses on improving the group robustness for VLM, as exemplified by standard benchmarks such as Waterbirds and CelebA, where each image is associated with a single class label. The proposed method does not assume that prompts are limited to object names; rather, it operates on any set of class-level textual descriptions (e.g., “a photo of a [class]” or attribute-augmented variants). To address more complex visual contexts, we have already included experiments on COCO-FP, a benchmark featuring multi-object scenes with distractor backgrounds. However, extending our framework to a full region-aware grounding pipeline—which would involve object detection, region selection, and localized prompt construction—would entail significant additional engineering changes. Such an extension lies beyond the scope of this paper, which aims to study robustness in *global image classification* under spurious correlations. We acknowledge this limitation in the revised manuscript and identify region-level prompting as a promising direction for future work.

---

> ### Author Response · Authors · 2025-12-10
> **Official Comment by Authors [4/4]**
>
> **The Computational Complexity**
>
> In the original manuscript, we only provided the asymptotic complexity of DoubleCCA:
>
> $O(K(d^2 + d^2_{\text{se}}) + d^3),$
>
> where $K$ is the number of augmented sentences, $d$ is the CLIP text dimension, and $d_{\text{se}}$ is the sentence embedding dimension. Note that all CCA computations are performed offline.During the inference, only a dot product operation between the merged text embedding W and image features is required. This operation has time complexity $O(|\mathcal{Y}|\cdot d)$, which matches that of the original CLIP.
>
> Following the reviewer’s suggestion, we provide quantitative results—including total inference time, CPU/GPU memory consumption, and FLOPs—on the full test sets of Waterbirds and CelebA across four backbone architectures we used in previous version. The results are summarized as follows (more details refer to the Appendix in our revised version):
>
> - FLOPs increase by approximately 10% (e.g., RN50: 4.1G → 4.51G; ViT-L/14: 81.1G → 89.21G);
> - Wall-clock inference time increases by only approximately 0.3–1.6%;
> - GPU/CPU memory increases by approximately 1–3%.
>
> These results show that DoubleCCA incurs negligible additional computational overhead during inference, with inference speed remaining virtually unchanged. This aligns with our complexity analysis and empirically validates the claim that DoubleCCA is computationally efficient.
>
> The results are shown in the following table:
> |  |  | Waterbirds |  |  |  | CelebA |  |  |  |
> | --- | --- | --- | --- | --- | --- | --- | --- | --- | --- |
> | Model |  | Time(s) | CPU(MB) | GPU(MB) | FLOPs(G) | Time(s) | CPU(MB) | GPU(MB) | FLOPs(G) |
> | RN50 | Origin | 40.67 | 3,355 | 1,699 | 4.1 | 251.67 | 7,282 | 1,695 | 4.1 |
> |  | Ours | 41.1 | 3,359 | 1,733 | 4.51 | 252.82 | 7,315 | 1,733 | 4.51 |
> | ViT-B/32 | Origin | 42.01 | 3,650 | 1,135 | 4.4 | 248.78 | 5,626 | 1,135 | 4.4 |
> |  | Ours | 42.69 | 3,712 | 1,173 | 4.84 | 250.25 | 5,635 | 1,173 | 4.84 |
> | ViT-B/16 | Origin | 51.97 | 3,589 | 1,865 | 17.6 | 246.49 | 5,672 | 1,865 | 17.6 |
> |  | Ours | 52.8 | 3,594 | 1,903 | 19.36 | 248.05 | 5,696 | 1,903 | 19.36 |
> | ViT-L/14 | Origin | 50.63 | 5,565 | 3,209 | 81.1 | 577.62 | 8,123 | 3,209 | 81.1 |
> |  | Ours | 51.13 | 5,622 | 3,243 | 89.21 | 579.27 | 8,165 | 3,243 | 89.21 |
>
> We have added the above analysis and new results in our revised manuscript.

---

### Review · Reviewer_xbMA · 2025-10-14

**Summary Of Contributions:**

The authors propose DoubleCCA, a two-stage CCA pipeline that fuses CLIP text embeddings with an auxiliary sentence-embedding model to improve group robustness (worst-group accuracy, robustness gap) in zero-shot classification. Stage-1 aligns CLIP and sentence embeddings into a shared space; Stage-2 “merges” predictors via another CCA and maps back for compatibility with CLIP image features. The method is plug-and-play (no model finetuning), and experiments on Waterbirds and CelebA report improved worst-group accuracy and reduced gaps across several CLIP backbones.

**Strengths**:
- Clear problem framing: Addresses group robustness for VLMs where text encoders may entangle semantics with spurious visual correlations. Provides an attribute bias score and t-SNE visualizations to motivate the approach.
- Simple, modular idea: Uses off-the-shelf CCA twice; no gradient updates to the backbone. Can be combined with PerceptionCLIP, contrastive adapters, etc.
- Empirical breadth across backbones: RN50, ViT-B/32, ViT-B/16, ViT-L/14, with consistent gains in many settings, especially on worst-group accuracy.
- Ablations: On CCA dimension, number of augmented sentences, single vs. double CCA, and alternative sentence encoders (HiT, SBERT, GTE, BART). These help isolate what matters.

**Weaknesses**:
- The paper reports +gender, +gender, age, +gender, age, and race for CelebA and even lists “white skin, dark skin, Asian, and others” as contextual attributes (Table 1 text). CelebA does not contain a race label by default. The paper does not explain how these labels are derived (manual annotation? external classifier? skin-tone proxy?), nor does it discuss ethical implications or label quality.
- Early sections argue prior methods rely on LLMs or external knowledge and face cost/generalization issues; however, the augmentation explicitly includes LLM-generated sentences (“like Qwen”) as a standard component to stabilize CCA (Alg. 1 step 1; §3.2.3; Fig. 8; Appendix E/Table 3). This weakens the stated motivation that the method avoids LLM reliance and extra cost.
- The method sometimes conditions prompts on group/context attributes (e.g., background for Waterbirds; gender/age/race for CelebA) following PerceptionCLIP. It is unclear whether at test time the model is allowed to use the ground-truth group attribute in the prompt (which would leak spurious information) or whether you enumerate multiple contexts and marginalize. Please spell out the exact inference protocol for every row in Tables 1–2 to ensure fairness and comparability.

**Additional Comments:**

NA

**Audience:**

Yes

**Audience Explanation:**

The paper directly addresses group robustness and bias mitigation in foundation models, topics that are central to TMLR’s stated interests in improving the reliability, fairness, and generalization of machine learning systems

**Claims And Evidence:**

Yes

**Claims Explanation:**

- Consistent worst-group gains on Waterbirds and CelebA across several CLIP backbones, with results summarized in Tables 1–2; the text explicitly frames the trade-off between average accuracy and worst-group robustness and shows improvements under both plain prompts and when integrated with PerceptionCLIP’s contextual prompts.

- Method clarity: the DoubleCCA procedure, its inference rule, and computational profile are specified (Alg. 1; Eq. 11; complexity), supporting reproducibility in principle.

- Ablations on augmentation show that LLM-generated sentences materially boost worst-group accuracy and that “full” augmentation works best (Table 3, CelebA/ViT-L/14).

- Qualitative + quantitative diagnostics (t-SNE + Attribute Bias Score/centroid distances) support the claim that auxiliary encoders are less sensitive to spurious attributes.

**Requested Changes:**

I would appreciate it if the author could address or elaborate on the areas of improvement.

---

> ### Author Response · Authors · 2025-12-10
> **Official Comment by Authors [1/4]**
>
> **Clarification on the Use of Race Attributes in CelebA**
>
> Thank you for bringing this to our attention. Indeed, the CelebA dataset does not include race annotations by default. We sincerely apologize for any confusion this may have caused and hereby clarify the origin and usage of these attributes.
>
> 1. **Origin & Replication**. We would like to clarify that the race attributes mentioned in our manuscript, such as “White,” “Black,” and “Asian”, are neither manual annotations we produced nor labels generated by an external classifier. It should be noted that the race attributes are used here for the purpose of conducting a strict and fair comparison with PerceptionCLIP (An et al., ICLR 2024), and thus we fully adhere to the list of contextual attributes defined in the open-source code of this paper.
> 2. **Technical Mechanism.** In our experimental setup (corresponding to the “+race” rows in Table 1), the race-related terms (e.g., “White,” “Black,” “Asian”) are used exclusively as prompt candidates during text embedding construction and do not serve as ground-truth labels for evaluation. Specifically, the model computes the similarity between the input image and a set of attribute-specific text descriptions during inference. It then dynamically infers the most likely attribute (e.g., race or gender) by treating the CLIP model itself as a *soft classifier*. This inferred attribute is subsequently injected as contextual information into the text encoder to aid the primary classification task (e.g., hair color prediction).
> 3. **Evaluation Metric**. It’s worth noting that the "Worst-group Accuracy" reported in Table 1 is still computed based on the attributes officially provided by CelebA, not based on race. Race information is used only as auxiliary input context and does not participate in evaluation grouping.
>
> Therefore, our method does not require CelebA to include race labels, nor have we performed any additional race annotation on the dataset. We will explicitly clarify this point in the revised manuscript.

---

> ### Author Response · Authors · 2025-12-10
> **Official Comment by Authors [2/4]**
>
> **The Problem of the LLM usage**
>
> We thank the reviewer for indicating this problem. We apologize for this conflict description in our previous manuscript. In fact, our goal is not to completely avoid using LLMs, but rather to minimize heavy reliance on interaction with LLM such as LLM-based synthesized datasets or dataset-specific prompt optimization, etc. Because these typically incur substantial computational or API costs, limiting the practicality of models in real-world systems.
>
> In contrast to prior work, DoubleCCA uses LLMs differently: we employ an LLM only once per class to generate a small set of auxiliary sentences that serve as a form of data augmentation. Specifically, for an evaluation dataset with *N* classes, we invoke an LLM (e.g., the Qwen series) once for each class to produce a few semantically plausible and dataset-agnostic extended descriptions. These generated sentences enrich the original class-level text prompts and thereby stabilize the optimization of CCA. Note that, we do not invoke the LLM during inference, so that the inference of our method is efficient on par with the original CLIP. On the other hand, our method also differs from approaches such as *Yang et al.* [1], which fully rely on complex synthetic pipelines that require LLMs to generate balanced text data for each dataset and subsequently train the learnable tokens on this synthetic data. Differently, our approach does not require complex, meticulously designed processes. It merely utilizes large models to assist in generating a small number of sentences, thereby enhancing textual diversity and stabilizing CCA optimization.
>
> More importantly, LLMs are an optional rather than required component in DoubleCCA. As the results shown in Section 4.4.3 and Appendix E (ranging from S-1 to S-4), we observe that using only the original prompt (S-1) already yields modest worst-group accuracy gains. In contrast, augmenting it with random characters alone (S-2) noticeably degrades performance. Adding LLM-generated descriptions (S-4) further improves robustness, and the full combination—original prompt + random characters + LLM-generated descriptions (S-5)—achieves the best performance and stability, which is why it is adopted as our main experimental configuration.
>
> In our revised version, we will make the following modifications accordingly to clarify the usage of LLM.
>
> (1) We will revise our introduction statement on the “limitations and costs of relying on LLMs” to clarify that our goal is to reduce reliance on LLM-generated data and complex prompt-tuning pipelines—not to avoid using LLMs altogether—thereby avoiding any misinterpretation that our method is entirely LLM-free.
>
> (2) We will explicitly clarify that LLMs are used only as an optional, offline data augmentation tool, and that the DoubleCCA framework itself is agnostic to the choice of LLM.
>
> (3) We will more prominently highlight in the main text the ablation results demonstrating that our method remains effective even when LLMs are not used or are used minimally (see Figure 8 and Table 4), ensuring full consistency between our motivation—to reduce reliance on complex LLMs—and the supporting empirical evidence.
>
> [1] Debiasing vison-language models with text-only training.

---

> ### Author Response · Authors · 2025-12-10
> **Official Comment by Authors [3/4]**
>
> **Clarification on the Inference Protocol**
>
> Thank you for raising this important concern. We clarify that in all our experiments, we do not use any ground-truth dataset attributes—such as background, gender, or age—to construct prompts. These group or contextual attributes are used solely during evaluation to compute worst-group accuracy and are never provided as input to the model.
>
> Specially, we provide additional details of our inference protocol as follows.
>
> 1. **Inference Protocol Based on PerceptionCLIP**
>
> In fact, we strictly adhered to PerceptionCLIP’s experimental setup to ensure a fair comparison. In Tables 1–2, we adopt its two-stage inference pipeline with attribute combinations such as “+background,” “+gender,” “+gender,age,” and “+gender,age,race,” without incorporating any additional information beyond what is specified in PerceptionCLIP. Here, we first introduce the details of the two-stage inference pipeline.
>
> (1) Stage 1: Attribute Inference
>
> Given a test image, we follow PerceptionCLIP’s implementation: CLIP computes similarity scores between the image and a set of predefined attribute templates—e.g., {“on grass,” “in the water,” “in a forest,” “with sky,” “on a beach,” …} for Waterbirds, and {“male”/“female,” “young”/“old,” “light skin”/“dark skin”/“Asian”/“other”} for CelebA. These scores yield an approximate posterior distribution$p(z \mid I)$, where *z* denotes contextual attributes such as background, gender, age, or  race. Critically, all such attributes are predicted by the model at inference time and do not rely on any ground-truth annotations from the dataset.
>
> (2) Stage 2: Conditional Classification and Aggregation
>
> For each candidate contextual attribute *z* (e.g., background for Waterbirds; gender and age for CelebA), we construct class-conditional prompts that incorporate context:
>
> - **Waterbirds**: “a photo of a {class} on {background}”
> - **CelebA**: “a photo of a {gender} {age} person with {hair color}”
>
> We then use CLIP (or CLIP + DoubleCCA) to compute the conditional class probability $p(y∣I,z)$. Following PerceptionCLIP, we perform a weighted aggregation to approximate marginalization over *z*, which uses the estimated posterior $p(z∣I)$ to obtain the final prediction $p(y∣I)=\sum_zp(y∣I,z)p(z∣I)$.
>
> Crucially, this entire process is unsupervised at inference time: no ground-truth attributes are used—only model-generated estimates.
>
> As noted in Section 4.2 of the original manuscript: *“We only replace the original prompt embedding with the merged text embedding W and adopt the same inference process as PerceptionCLIP.”*
>
> 2. **Inference Protocols Across Experimental Settings**
>
> First, we provide a row-by-row breakdown of the inference protocol for the main entries in Tables 1–2, explicitly indicating whether ground-truth group labels are used. In all cases, no ground-truth group information is used at any stage of inference. The details are described in the following Table.
>
> | Row / Setting                                                                 | Inference Protocol Summary                                                                                                                                                     | Uses GT Group Labels? |
> |------------------------------------------------------------------------------|--------------------------------------------------------------------------------------------------------------------------------------------------------------------------------|------------------------|
> | CLIP                                                                         | Standard zero-shot: uses template "a photo of a {class}", predicts $\hat{y} = \arg\max_y \cos(\Phi_t(T_y), \Phi_v(I))$.                                                        | No                     |
> | Ours (DoubleCCA)                                                             | Same zero-shot inference pipeline as CLIP, except all $\Phi_t(T_y)$ are replaced with our merged embeddings $W_y$: $\hat{y} = \arg\max_y \cos(W_y, \Phi_v(I))$.                | No                     |
> | +background / +gender / +gender,age / +gender,age,race (PerceptionCLIP)       | Follows exactly the two-stage pipeline of PerceptionCLIP: first uses CLIP to predict the context distribution $p(z \mid I)$, then performs conditional classification under each candidate context and aggregates. | No                     |
> | Ours + background / Ours + gender / Ours + gender,age / Ours + gender,age,race | Based on the above PerceptionCLIP pipeline, only replacing all text embeddings $\Phi_t(\cdot)$ with our pre-computed merged embeddings $W$; attributes are still automatically predicted and marginalized by the model. | No                     |

---

> ### Author Response · Authors · 2025-12-10
> **Official Comment by Authors [4/4]**
>
> Second, we present the details of inference protocol for each dataset.
>
> (1) On Waterbirds
>
> - **CLIP** and **Ours**: Use only class-name prompts (e.g., “a photo of a [class]”) without incorporating any background information.
> - **+background (PerceptionCLIP)**: The background attribute is first predicted by CLIP using a set of background-specific attribute templates (e.g., “on grass,” “in the water”). The top predicted background is then injected into the class prompt (e.g., “a photo of a [class] on grass”) to perform conditional classification. This two-stage inference pipeline exactly follows the original PerceptionCLIP protocol.
> - **Ours + background**: We replace all text embeddings in the above PerceptionCLIP pipeline with the merged embedding **W** produced by DoubleCCA. All other steps—including background prediction and prompt composition—remain unchanged.
>
> (2) On CelebA
>
> - **CLIP** and **Ours**: Use only hair-color class-name prompts (e.g., “a photo of a person with blond hair”), without incorporating gender, age, or race.
> - **+gender / +gender,age / +gender,age,race** (**PerceptionCLIP**): These settings treat gender, gender+age, and gender+age+race as contextual factors *z*. At test time, all such attributes are predicted by the model from the input image using CLIP-based attribute templates and dynamically inserted into the prompt (e.g., “a photo of a young female person with blond hair”). No ground-truth attributes are used during inference; they are employed only during evaluation to define groups for computing worst-group and average metrics.
> - **Ours + …**: For each of the above configurations, we adopt exactly the same attribute vocabulary, prompt templates, and inference protocol as the corresponding PerceptionCLIP variant. The only difference lies in the text representation: we replace CLIP’s original text embeddings with the merged embedding **W** produced by DoubleCCA.
>
> Consequently, across all rows involving contextual attributes, our method maintains alignment with PerceptionCLIP’s inference pipeline. The sole modification is the underlying text embedding—ensuring that performance differences stem solely from representational quality, not protocol discrepancies. This guarantees a fair and controlled comparison.
>
> We will include all these descriptions in our revised manuscript.

---

### Review · Reviewer_gN8Q · 2025-11-10

**Summary Of Contributions:**

This paper proposes DoubleCCA, a plug-and-play framework to improve group robustness of vision–language foundation models. The key idea is to enrich CLIP’s text embeddings with an auxiliary sentence embedding model by employing Canonical Correlation Analysis (CCA) twice. The first CCA aligns CLIP text embeddings and auxiliary sentence embeddings into a shared space, while the second CCA reconstructs invariant features for compatibility with visual features. Experiments on Waterbirds and CelebA show consistent improvements in worst-group accuracy and reduced robustness gaps.

Strength
- The empirical analyses convincingly show that auxiliary sentence encoders produce more invariant text representations.
- It is easy to integrate with existing CLIP pipelines.

Weakness
- Using CCA to fuse multi-view embeddings is established, so the contribution seems like an application-specific design (a second CCA to reconstruct into CLIP space). The authors should strengthen the description of the method's innovativeness by comparing it with methods in the field and papers related to CCA.
- Using text embeddings as a “proxy” for image features in the second CCA is heuristic; stronger justification or an empirical sanity check (e.g., alignment metrics w.r.t. actual image features) would strengthen the plausibility of this design.
- The robustness depends on the data augmentation, which uses LLM-generated sentences alongside random sequences. Since the paper initially critiques reliance on external knowledge/LLMs, there is a potential conflict here.
- The authors claim that their method is "computationally efficient". However, the paper provides only a theoretical analysis of its complexity. A more convincing demonstration would include a discussion of the method’s complexity in comparison with other approaches and empirical experiments that quantify and compare the computational cost.

**Audience:**

Yes

**Audience Explanation:**

The paper’s focus on enhancing group robustness in vision–language models aligns with TMLR’s scope, and audience in this field would be interested in its findings.

**Claims And Evidence:**

Yes

**Claims Explanation:**

The claims are mostly supported by accurate and convincing analyses and  experiments.

**Requested Changes:**

Please see the weakness

---

> ### Author Response · Authors · 2025-12-10
> **Official Comment by Authors [1/5]**
>
> **On Methodological Innovation**
>
> Thank you for raising this concern. We clarify the motivation behind our DoubleCCA design—which applies the classical CCA technique twice—through four perspectives detailed below.
>
> 1. **New Problem Definition and Application Scenario**
>
> Previous CCA-related works have mainly focused on multi-view feature embeddings or model merging. For instance, *Dhillon et al.* [1] use CCA to embed word embedding features into a common feature space to enhance representation ability. *Horoi et al.* [2] propose using CCA to merge parameters and representations of deep models with identical architectures, aiming to improve representation quality or ensemble performance.
>
> Although DoubleCCA applies the classical CCA technique twice, our target is to mitigate spurious bias in CLIP's text encoder with respect to group attributes, thereby improving worst-group accuracy in zero-shot classification. In Section 3 and the Appendix, we demonstrate through attribute bias scores and t-SNE visualizations that CLIP text embeddings exhibit significant shifts across background/gender attributes, while sentence embedding models like HiT/BERT are more stable.
>
> Therefore, DoubleCCA is specifically designed to address this text-side spurious bias phenomenon.
>
> 2. **Rationale for the Two-Stage Design**
>
> In contrast to prior works that employ CCA techniques, the two CCA stages in our method serve fundamentally different roles.
>
> (1) The first CCA aims to align CLIP text embeddings with auxiliary sentence embeddings in a shared semantic subspace that is both richer in semantics and well-aligned.
>
> (2) The second CCA performs optimal linear fusion in two predictor spaces and projects the result back into the original CLIP text embedding space to preserve compatibility with the image encoder.
>
> To the best of our knowledge, this is the first work to apply this classical technique to enhance group robustness in CLIP-based models. As shown in Figure 7(c), our ablation studies further demonstrate the importance of the two-stage CCA design. While the first-stage CCA alone yields modest performance gains, it is insufficient to achieve optimal results. In contrast, the full two-stage formulation substantially improves worst-group performance, highlighting that the two stages are both necessary and complementary for this task. This directly refutes the notion that the second CCA is a redundant design; rather, it is a crucial step to ensure compatibility with CLIP’s visual embedding space.
>
> 3. **Computational Efficiency of DoubleCCA**
>
> Furthermore, our method employs the classical CCA technique, which admits a closed-form solution and thus enables efficient optimization of the objective. Therefore, during this revision, our new experiments verify that DoubleCCA achieves consistent performance improvements with almost no increase in inference time compared to original CLIP. As shown in our revision, DoubleCCA increases FLOPs by approximately 10%, while the actual wall-clock inference time rises by only about 1% and GPU memory usage grows by roughly 1–3%. This means that DoubleCCA achieves meaningful fairness improvements while preserving the original inference efficiency of CLIP. This property makes it well-suited for large-scale deployment scenarios.
>
> 4. **Discussion with Previous works.**
>
> (1) The first CCA step is similar to previous CCA works focused on multi-view embedding, which aims to fuse the CLIP text embedding with auxiliary sentence embedding in a low-rank embedding space. However, our method differs by incorporating a second CCA step, which is designed to recover the embedding space back to the original CLIP feature space. Our experiment 4.4.2 validates the necessity of this second step.
>
> (2) Unlike Horoi et al. [2], which requires extracting and fusing multi-layer features from within trained networks, our method operates on class-level text embeddings via closed-form CCA transformations, without accessing training images or modifying/fine-tuning the backbone. Moreover, it is a flexible, plug-and-play module that can be seamlessly integrated with existing group-robustness methods such as PerceptionCLIP and Contrastive Adapter (see Table 2 and Figure 5).
>
> [1] Multi-View Learning of Word Embeddings via CCA. NeurIPS 2011.
> [2] Harmony in diversity: Merging neural networks with canonical correlation analysis. ICML 2024.

---

> ### Author Response · Authors · 2025-12-10
> **Official Comment by Authors [2/5]**
>
> 5. **Summary**
>
> We believe the innovations of DoubleCCA are reflected in the following aspects:
>
> (1) DoubleCCA is the first method to explicitly combine auxiliary sentence embedding models with a two-stage CCA framework specifically for improving group robustness in VLMs;
>
> (2) It introduces a novel *text-side align-then-merge-back* framework and demonstrates that the second CCA stage is crucial for robustness gains;
>
> (3) It achieves debiased representations that remain compatible with CLIP’s visual embedding space, while strictly freezing the backbone and using only class-level text prompts;
>
> (4) It inherits the computational efficiency of classical CCA’s closed-form solution, thereby incurring almost no increase in the inference speed of the CLIP model.
>
> In our revision, we make three key updates. First, we add a new discussion comparing our method with previous works such as [1,2]. Second, we highlight the material about "Complementary Encoders" and information-theoretic motivation from Appendix D in the main text. Third, we provide a more detailed analysis of computational overhead in the appendix, including comparative results on FLOPs, wall-clock inference time, and GPU memory usage.
>
> [1] Multi-View Learning of Word Embeddings via CCA. NeurIPS 2011.
> [2] Harmony in diversity: Merging neural networks with canonical correlation analysis. ICML 2024.

---

> ### Author Response · Authors · 2025-12-10
> **Official Comment by Authors [3/5]**
>
> **On the Rationale for Using Text as an Image Proxy**
>
> Thank you for raising this interesting point. We explain our use of text as a proxy for images below, with supplementary clarification from three perspectives: theoretical, empirical, and design trade-offs.
>
> 1. **Theoretical Analysis**
>
> We argue that using text as a proxy for images benefits from CLIP's alignment ability, which achieves the best available approximation.
>
> In our work, we focus on zero-shot inference; therefore, we do not access any training or validation images from the target dataset and cannot directly leverage image features from the target domain. To mitigate this limitation, we exploit CLIP’s inherent text–visual alignment: for semantically aligned text–image pairs (*T*,*I*), the expected class-conditional text and image embeddings are approximately equal, i.e.,
>
>  $\mathbb{E}[\Phi_t(T)|y] \approx \mathbb{E}[\Phi_v(I)|y]$
>
> as formalized in Proposition 1 (Appendix D). Consequently, under the constraint of no access to target-domain images, class-level text embeddings serve as our best available proxy for the image feature distribution of each class.
>
> 2. **Empirical Validation**
>
> To validate this proxy assumption, we conduct a new ablation study in a setting where labeled target-domain image data are available, replacing the class-level text embeddings with real image features. The results is reported in the following table:
>
> | Proxy Type | Waterbirds Avg | Waterbirds Worst | CelebA Avg | CelebA Worst |
> | --- | --- | --- | --- | --- |
> | Text Embedding (ours) | 89.55 | 62.50 | 85.35 | 83.00 |
> | Real Image Features | 91.93 | 64.29 | 88.68 | 84.35 |
>
> We observe that using features extracted from real images indeed yields better performance, improving worst-group accuracy by 1–2% on average compared to the text-based proxy. This result empirically validates our core assumption that class-level text embeddings serve as a reasonable proxy for target-domain image features under zero-shot setting. Thus, we include ~~the~~ this new experiment and analysis in the Appendix of the revised manuscript.
>
> 3. **Design Trade-offs**
>
> Finally, we conclude that this design is an intentional trade-off.
>
> - Maintain a fully zero-shot DoubleCCA without requiring target domain images to facilitate rapid transfer to new datasets.
> - Compared to methods that use real image features, performance is slightly lower but remains competitive to a certain extent.
> - In real-world applications, where privacy or data acquisition is restricted, the text proxy offers greater engineering feasibility.

---

> ### Author Response · Authors · 2025-12-10
> **Official Comment by Authors [4/5]**
>
> **Clarification on LLM Usage**
>
> We thank the reviewer for indicating this problem. We apologize for this conflict description in our previous manuscript. In fact, our goal is not to completely avoid using LLMs, but rather to minimize heavy reliance on interaction with LLM such as LLM-based synthesized datasets or dataset-specific prompt optimization, etc. Because these typically incur substantial computational or API costs, limiting the practicality of models in real-world systems.
>
> In contrast to prior work, DoubleCCA uses LLMs differently: we employ an LLM only once per class to generate a small set of auxiliary sentences that serve as a form of data augmentation. Specifically, for an evaluation dataset with *N* classes, we invoke an LLM (e.g., the Qwen series) once for each class to produce a few semantically plausible and dataset-agnostic extended descriptions. These generated sentences enrich the original class-level text prompts and thereby stabilize the optimization of CCA. Note that, we do not invoke the LLM during inference, so that the inference of our method is efficient on par with the original CLIP. On the other hand, our method also differs from approaches such as *Yang et al.* [1], which fully rely on complex synthetic pipelines that require LLMs to generate balanced text data for each dataset and subsequently train the learnable tokens on this synthetic data. Differently, our approach does not require complex, meticulously designed processes. It merely utilizes large models to assist in generating a small number of sentences, thereby enhancing textual diversity and stabilizing CCA optimization.
>
> More importantly, LLMs are an optional rather than required component in DoubleCCA. As the results shown in Section 4.4.3 and Appendix E (ranging from S-1 to S-4), we observe that using only the original prompt (S-1) already yields modest worst-group accuracy gains. In contrast, augmenting it with random characters alone (S-2) noticeably degrades performance. Adding LLM-generated descriptions (S-4) further improves robustness, and the full combination—original prompt + random characters + LLM-generated descriptions (S-5)—achieves the best performance and stability, which is why it is adopted as our main experimental configuration.
>
> In our revised version, we will make the following modifications accordingly to clarify the usage of LLM.
>
> (1) We will revise our introduction statement on the “limitations and costs of relying on LLMs” to clarify that our goal is to reduce reliance on LLM-generated data and complex prompt-tuning pipelines—not to avoid using LLMs altogether—thereby avoiding any misinterpretation that our method is entirely LLM-free.
>
> (2) We will explicitly clarify that LLMs are used only as an optional, offline data augmentation tool, and that the DoubleCCA framework itself is agnostic to the choice of LLM.
>
> (3) We will more prominently highlight in the main text the ablation results demonstrating that our method remains effective even when LLMs are not used or are used minimally (see Figure 8 and Table 4), ensuring full consistency between our motivation—to reduce reliance on complex LLMs—and the supporting empirical evidence.
>
> [1] Debiasing vison-language models with text-only training.

---

> ### Author Response · Authors · 2025-12-10
> **Official Comment by Authors [5/5]**
>
> **The Computational Complexity**
>
> In the original manuscript, we only provided the asymptotic complexity of DoubleCCA:
>
> $O(K(d^2 + d^2_{\text{se}}) + d^3),$
>
> where K is the number of augmented sentences, d is the CLIP text dimension, and $d_{\text{se}}$ is the sentence embedding dimension. Note that all CCA computations are performed offline.During the inference, only a dot product operation between the merged text embedding W and image features is required. This operation has time complexity $O(|\mathcal{Y}|\cdot d)$, which matches that of the original CLIP.
>
> Following the reviewer’s suggestion, we provide quantitative results—including total inference time, CPU/GPU memory consumption, and FLOPs—on the full test sets of Waterbirds and CelebA across four backbone architectures we used in previous version. The results are summarized as follows (more details refer to the Appendix in our revised version):
>
> - FLOPs increase by approximately 10% (e.g., RN50: 4.1G → 4.51G; ViT-L/14: 81.1G → 89.21G);
> - Wall-clock inference time increases by only approximately 0.3–1.6%;
> - GPU/CPU memory increases by approximately 1–3%.
>
> These results show that DoubleCCA incurs negligible additional computational overhead during inference, with inference speed remaining virtually unchanged. This aligns with our complexity analysis and empirically validates the claim that DoubleCCA is computationally efficient.
>
> The results are shown in the following table:
> |  |  | Waterbirds |  |  |  | CelebA |  |  |  |
> | --- | --- | --- | --- | --- | --- | --- | --- | --- | --- |
> | Model |  | Time(s) | CPU(MB) | GPU(MB) | FLOPs(G) | Time(s) | CPU(MB) | GPU(MB) | FLOPs(G) |
> | RN50 | Origin | 40.67 | 3,355 | 1,699 | 4.1 | 251.67 | 7,282 | 1,695 | 4.1 |
> |  | Ours | 41.1 | 3,359 | 1,733 | 4.51 | 252.82 | 7,315 | 1,733 | 4.51 |
> | ViT-B/32 | Origin | 42.01 | 3,650 | 1,135 | 4.4 | 248.78 | 5,626 | 1,135 | 4.4 |
> |  | Ours | 42.69 | 3,712 | 1,173 | 4.84 | 250.25 | 5,635 | 1,173 | 4.84 |
> | ViT-B/16 | Origin | 51.97 | 3,589 | 1,865 | 17.6 | 246.49 | 5,672 | 1,865 | 17.6 |
> |  | Ours | 52.8 | 3,594 | 1,903 | 19.36 | 248.05 | 5,696 | 1,903 | 19.36 |
> | ViT-L/14 | Origin | 50.63 | 5,565 | 3,209 | 81.1 | 577.62 | 8,123 | 3,209 | 81.1 |
> |  | Ours | 51.13 | 5,622 | 3,243 | 89.21 | 579.27 | 8,165 | 3,243 | 89.21 |
>
> We have added the above analysis and new results in our revised manuscript.

---

### Decision · Action_Editor_MQgo · 2025-12-22

**Recommendation:** Accept as is

**Audience:**

Yes

**Audience Explanation:**

Since this work is about improving CLIP classification by embedding more semantic context, there will be TMLR's audience interested in knowing the findings of this paper.

**Claims And Evidence:**

Yes

**Claims Explanation:**

This paper proposes DoubleCCA, a two-stage CCA pipeline that fuses CLIP text embeddings with an auxiliary sentence-embedding model to improve group robustness (e.g., worst-group accuracy and robustness gap) in zero-shot classification. After the author rebuttal, the paper received two Leaning Accept and one Reject recommendations.

On the positive side, all reviewers agreed that (1) the proposed method is simple, modular, and can be used as a plug-in with clear intuition, and (2) the ablation studies are comprehensive.

Reviewers raised questions primarily regarding (1) the need for additional benchmark evaluations in out-of-distribution (OOD) settings and more complex scenarios, and (2) a more thorough discussion and analysis of computational costs, including text-embedding latency, FLOPs, and GPU/CPU memory usage for embedding generation and CCA at test time. The authors provided detailed responses to these concerns during the rebuttal.

For the reviewer who issued the reject recommendation, the justification suggests that the reviewer did not see responses to his feedback. This may be because the recommendation was submitted around the same time as the authors’ rebuttal.

Overall, I think the rebuttal added non-trivial experiments that make the paper more convincing.